# Beyond Scale: the Diversity Coefficient as a Data Quality Metric Demonstrates LLMs are Pre-trained on Formally Diverse Data

## Abstract

Current trends to pre-train capable Large Language Models (LLMs) mostly focus on scaling of model and dataset size. However, the *quality* of pre-training data is an important factor for training powerful LLMs, yet it is a nebulous concept that has not been fully characterized. Therefore, we use the recently proposed Task2Vec diversity coefficient to ground and understand formal aspects of data quality, to go beyond scale alone. Specifically, we measure the diversity coefficient of publicly available pre-training datasets to demonstrate that their formal diversity is high when compared to theoretical lower and upper bounds. In addition, to build confidence in the diversity coefficient, we conduct interpretability experiments and find that the coefficient aligns with intuitive properties of diversity, e.g., it increases as the number of latent concepts increases. We conclude the diversity coefficient is reliable, show it's high for publicly available LLM datasets, and conjecture it can be used to build useful diverse datasets for LLMs.

## 1 Introduction

Current trends in pre-training Large Language Models (LLMs) tend to concentrate on model and dataset size scaling Chowdhery et al. (2022); Nostalgebraist (2022); OpenAI (2023); Google (2023). Therefore, vast amounts of effort have been invested in understanding neural scaling laws – the power-law relationship between the loss of deep artificial networks and the *size* of the pre-training dataset and model for a fixed compute budget (Hestness et al., 2017; Rosenfeld et al., 2019; Henighan et al., 2020; Kaplan et al., 2020; Gordon et al., 2021; Hernandez et al., 2021; Jones, 2021; Zhai et al., 2022; Hoffmann et al., 2022; Clark et al., 2022; Neumann & Gros, 2022). In addition, recent work focuses on training a fixed model but using *more* tokens (Touvron et al., 2023). However, the effectiveness of these systems also fundamentally relies on the quality Longpre et al. (2023) and coverage of the pre-training data Hashimoto (2021); David et al. (2010) and not only the *size*. Unfortunately, data quality and coverage David et al. (2010) are often overlooked or discussed in vague and imprecise ways Longpre et al. (2023). Hence, we propose to ground the discussion of data quality through the diversity coefficient Miranda et al. (2022a), a data coverage metric that moves beyond scale alone. We extend the diversity coefficient to formally quantify data diversity of publicly available datasets and discover that LLMs are pre-trained on formally diverse data. We demonstrate the diversity coefficient is *high* for these pre-training datasets by comparing their formal diversity to the non-vacuous conceptually well-motivated lower and upper bounds of the diversity coefficient. In addition, to instill confidence in the usage of the diversity coefficient, we assess the interpretability of the coefficient as it relates to intuitive and expected properties of such a diversity metric. Concretely, we demonstrate:

1. The diversity coefficient increases as one concatenates more pre-training datasets of different sources.
2. We show the task embedding distances used in the diversity coefficient groups in a meaningful way, reflecting the conceptual and semantic information humans expect.
3. Using the Generative IN-Context Learning (GINC) Xie et al. (2021) dataset, we show that as the number of latent concepts[1] increases the diversity coefficient increases.

---

[1]Latent concepts represent document-level features such as semantics, structure, and style Xie et al. (2021).

4. We show that a larger, more diverse vocabulary leads to a higher diversity coefficient in the Generative IN-Context Learning (GINC) Xie et al. (2021) dataset.

Our key **contributions** are:

1. A paradigm shift beyond dataset scale to a data-centric machine learning perspective through a formal data quality metric – the diversity coefficient.

2. We advance discussions on data quality by measuring an aspect of quality – data diversity – using the diversity coefficient.

3. We further validate the diversity coefficient by demonstrating its interpretability and correlation with intuitive diversity properties aligned with human intuitions, e.g., the coefficient increases as more datasets are concatenated, the number of latent concepts increases, and a richer vocabulary is used.

4. We formally demonstrate the high diversity of public datasets for LLM pre-training is *high* using well-motivated lower and upper bounds.

5. Lastly, for ease of use of our method, we also study properties of different parameters for computing the formal diversity and therefore provide practitioners with simpler ways to evaluate the diversity coefficient.

Therefore, we conclude the diversity coefficient is reliable, and conjecture the diversity coefficient can be used to build quality diverse datasets for capable LLMs. In doing so, we hope this work inspires more systematic and effective techniques for dataset design beyond simply increasing the number of data points, sequences, or tokens.

## 2 METHODS

### 2.1 TASK2VEC EMBEDDINGS FOR SEQUENCE DATA

We use the Task2Vec diversity coefficient (Miranda et al., 2022a) to compute the formal diversity of a dataset, echoing the approach of (Nguyen et al., 2019) in using Task2Vec to characterize data. We choose Task2Vec embeddings since they have been shown to be effective "informational fingerprints" of embedded data by, at a high level, identifying which model weights are important in performing prediction on a given batch of text sequences (Achille et al., 2019) (see Appendix A for further explanation). The first step is to compute Task2Vec (vectorial) embeddings of a batch of sequences. The original Task2Vec method (Achille et al., 2019) embeds data (e.g. few-shot learning task) using the diagonal entries of the Fisher Information Matrix (FIM) that result from (partially) fine-tuning the final layer of a fixed neural network (also called a *probe network*) to solve the current task (or batch), similar in nature to (Edwards & Storkey, 2017) with probe-network based data analysis. We implement this framework by fine-tuning GPT-2 Radford et al. (2019) to predict the next token for each sequence in the current batch $B$, then compute the FIM as follows:

$$\hat{F}_B = \mathbb{E}_{x,t,\hat{x}_t} \nabla_w \log \hat{p}_w(\hat{x}_t | x_{t-1:1}) \nabla_w \log \hat{p}_w(\hat{x}_t | x_{t-1:1})^\top$$

The Task2Vec embedding $\vec{f}_B$ is the diagonal ($Diag$) of the FIM $\vec{f}_B = Diag(F_B)$, where $x$ is a sequence of length $T_x$ sampled from a batch $B$ i.e. $x \in B$, $\hat{x}$ is a sequence of tokens sampled from the fine-tuned probe network $f_w$ (with weights $w$) conditioned on the real sequence $x$ i.e. $\hat{x} \sim \hat{p}_w(\hat{x}_t \mid x_{t-1:1})$, $t$ indicates taking the average log loss over the sequence length.

### 2.2 DIVERSITY COEFFICIENT COMPUTATION FOR NATURAL LANGUAGE DATASETS

#### 2.2.1 DEFINITION OF THE DIVERSITY COEFFICIENT

Using our extension of Task2Vec for sequence data, we explain how to compute the Task2Vec diversity coefficient (Miranda et al., 2022a) for natural language datasets using GPT-2 as a probe network. We compute the Task2Vec diversity coefficient as the expected cosine distance $d$ between pairs of Task2Vec embeddings of batches:

$$\hat{d}\mathrm{iv}(D) = \mathbb{E}_{B_1,B_2 \sim D} d(\vec{f}_{B_1}, \vec{f}_{B_2})$$

where $D$ is the natural language dataset from which we sample batches $B_1, B_2$, and $\vec{f}_{B_i}$ is the Task2Vec embedding of a batch $B_i$ using the diagonal of the FIM matrix $\hat{F}_{B_i}$. In this setting, if $D$ is a *union* of datasets (also known as *interleaved*), then a batch contains sequences from both datasets according to some specified data mixture.

By measuring the distance between FIMs, the diversity coefficient captures the average intrinsic variability of batches in the underlying data distribution as a proxy for data coverage or information contained in the dataset. Another interpretation is that dataset diversity reflects how different batches are from each other. See Figure 5 in Appendix F for an illustration of this process.

### 2.2.2 DEFINITION OF THE CROSS DIVERSITY COEFFICIENT

The cross diversity coefficient computes the expected cosine distances of (Task2Vec) embeddings of batches by sampling each batch of sequences from only one of the two datasets, without any interleaving or mixing of the sequences between datasets. In other words, each batch will only contain sequences from *one* of the sub-datasets, not both:

$$\hat{d}iv(D_1, D_2) = \mathbb{E}_{B_1 \sim D_1, B_2 \sim D_2} d(\vec{f}_{B_1}, \vec{f}_{B_2})$$

In this work, we use the term *concatenated* when the sequences in each batch come only from a single data set. We introduce these two definitions (diversity and cross diversity) to show our results hold with respect to two intuitive and logical ways to define data diversity (details in Appendix D).

### 2.2.3 BACKBONE USED TO COMPUTE THE DIVERSITY COEFFICIENT

To compute Task2Vec embeddings, we use GPT-2 Radford et al. (2019) pre-trained on the English language as the probe network $f_w$. Following Task2Vec (Achille et al., 2019), we fine-tune only the final layer (a language modeling head) on each batch since it's the only tested method for computing Task2Vec embeddings Achille et al. (2019); Miranda et al. (2022a; 2023), e.g. it's not known if the intuitive properties observed in (Achille et al., 2019) hold without fine-tuning the backbone. See Figure 5 for a visual of the diversity coefficient computation pipeline.

### 2.2.4 RECIPE FOR ESTABLISHING IF A DIVERSITY COEFFICIENT IS HIGH VIA THE CONCEPTUAL LOWER AND UPPER BOUNDS

To establish if a diversity coefficient $\hat{d}iv(D)$ of a dataset $D$ is high (or low), we use two conceptually well-motivated reference values. We call them the lower and upper bounds of the diversity coefficient. To understand the lower bound, consider a dataset constructed by sampling with most of the probability mass concentrated on some arbitrary token. This is a good candidate for a dataset with minimum diversity. To understand the upper bound, consider the other extreme: a dataset constructed by sampling any token uniformly at random given a fixed vocabulary (in our case, the GPT-2 tokenizer vocabulary) is a good candidate to create a dataset with maximum diversity.

Therefore, we measure a conceptual lower bound on a dataset with a vocabulary size of 2: `<eos>` token and a randomly selected non-special token from the GPT-2 tokenizer vocabulary. The `<eos>` token was assigned a probability weight of $1/\{\text{GPT-2 vocab size}\}$. The non-special token was assigned the remaining weight. Similarly, a high or maximum diversity dataset would consist of random sequences of all possible tokens, with no underlying order to semantics, formatting, etc. The upper bound of the diversity coefficient was therefore measured on a synthetic dataset with an equal probability of occurrence assigned to all tokens in the GPT-2 tokenizer vocabulary.

## 3 EXPERIMENTS & RESULTS

### 3.1 DIVERSITY COEFFICIENTS OF PRE-TRAINING DATA SHOWS LLMS ARE PRE-TRAINED ON FORMALLY HIGHLY DIVERSE DATA

**Experiments:** We evaluate the diversity coefficient (described in section 2.2.1) and cross diversity coefficient (described in section 2.2.2) of ten publicly available LLM pre-training datasets (described in section B). We also compute the diversity and cross diversity coefficients of two concatenated datasets: 1) C4 and WikiText-103, and 2) five sub-datasets of The Pile: Pile-CC, HackerNews,

Table 1: **Diversity coefficients of LLM pre-training datasets are 2.7-4.76 times higher than the conceptual lower bound and more than half that of the upper bound. Cross diversity coefficients of LLM pre-training datasets are 3-5 times higher than the conceptual lower bound and more than half that of the upper bound. Overall, any strategy of concatenating datasets increases the cross diversity coefficient**. For the diversity coefficient, batches of text sequences were sampled from a mixed (i.e. interleaved) pool of sequences from all sub-datasets. Thus, sequences from both sub-datasets are present in the same batch at a rate dictated by the data mixture. Mix1 stands for a data mixture with ratio 3:1 (i.e., 0.75 to 0.25) for the corresponding combined data sets. Mix2 stands for a data mixture according to LLaMA v1 (i.e., 0.77, 0.23) for the corresponding combined data sets (see Appendix I.6 for details). Note, cross diversity does *not* mix datasets when computing the corresponding coefficient. Instead, we sample batches entirely from one of the sub-datasets; the distance between batches is then used to compute the cross diversity (see Appendix D for explanation).

| DATASET | DIVERSITY COEFF. | CROSS DIVERSITY COEFF. |
|---|---|---|
| LOWER BOUND (LB) | **0.0525** $\pm$ 3.41E-4 | (SAME AS LEFT) |
| NIH EXPORTER | 0.15 $\pm$ 3.218E-5 | (SAME AS LEFT) |
| USPTO | 0.1582 $\pm$ 4.09E-5 | (SAME AS LEFT) |
| PUBMED ABSTRACTS | 0.168 $\pm$ 2.63E-5 | (SAME AS LEFT) |
| HACKERNEWS | 0.201 $\pm$ 4.52E-5 | (SAME AS LEFT) |
| WIKITEXT-103 | 0.2140 $\pm$ 7.93E-5 | (SAME AS LEFT) |
| COMBINATION OF FIVE DATASETS (MIX2) | **0.217** $\pm$ 9.81E-4 | **0.2939** $\pm$ 2.03E-4 |
| SLIMPAJAMA | 0.221 $\pm$ 9.97E-4 | (SAME AS LEFT) |
| OPENWEBTEXT | 0.222 $\pm$ 1.00E-3 | (SAME AS LEFT) |
| C4 AND WIKITEXT-103 (MIX1) | **0.235** $\pm$ 1.04E-3 | **0.2711** $\pm$ 3.22E-4 |
| C4 | 0.2374 $\pm$ 2.785E-5 | (SAME AS LEFT) |
| THE PILE | 0.2463 $\pm$ 3.034E-5 | (SAME AS LEFT) |
| PILE-CC | **0.2497** $\pm$ 3.41E-5 | (SAME AS LEFT) |
| UPPER BOUND (UB) | **0.4037** $\pm$ 1.932E-5 | (SAME AS LEFT) |

NIH ExPorter, PubMed, and USPTO (Appendix I.4). In addition, we compute our conceptually well-motivated lower and upper bounds on the diversity coefficient (section 2.2.4).

**Results:** Table 1 reports the aforementioned diversity and cross diversity coefficients. The key observations from our results are:

- The cross diversity coefficients of pre-training datasets tend to be **3-5 times greater than the theoretical lower bound and, on average, half the upper bound.** Prominently, C4, The Pile, and Pile-CC exhibit the highest diversity coefficients (0.23 - 0.25). This aligns with intuition, as these are very large, web-crawl-based, internet-scale datasets.

- The diversity coefficients of pre-training datasets tend to be **2.7-4.76 times greater than the theoretical lower bound and, on average, half the upper bound.** As expected, this is slightly lower than the cross diversity coefficient because the cross diversity coefficient compares batch embeddings from different, disjoint datasets.

- Three sub-datasets of The Pile—NIH ExPorter, PubMed Abstracts, and USPTO—show relatively low diversity (0.15 - 0.17), less than half of the upper bound (0.4). These datasets are curated from specialized fields with regular formats and topics, which may account for this observation.

- However, we observe that Pile-CC and HackerNews have higher diversity (.20 - .25), which may be attributed to their coverage of a broad range of topics.

## 3.2 CONCATENATION OF DATASETS OF DIFFERENT SOURCES PRODUCES HIGHER MEASURED DIVERSITY

**Experiments:** To show that the concatenation of different datasets produces high diversity datasets, we measure the cross diversity coefficient of C4 plus WikiText-103, as well as of the five sub-datasets of The Pile in Table 1. To understand the source of this increased diversity, we plot in Figure 1 the Task2Vec (cosine) distances between batches from individual datasets and distances of batches from the different datasets.

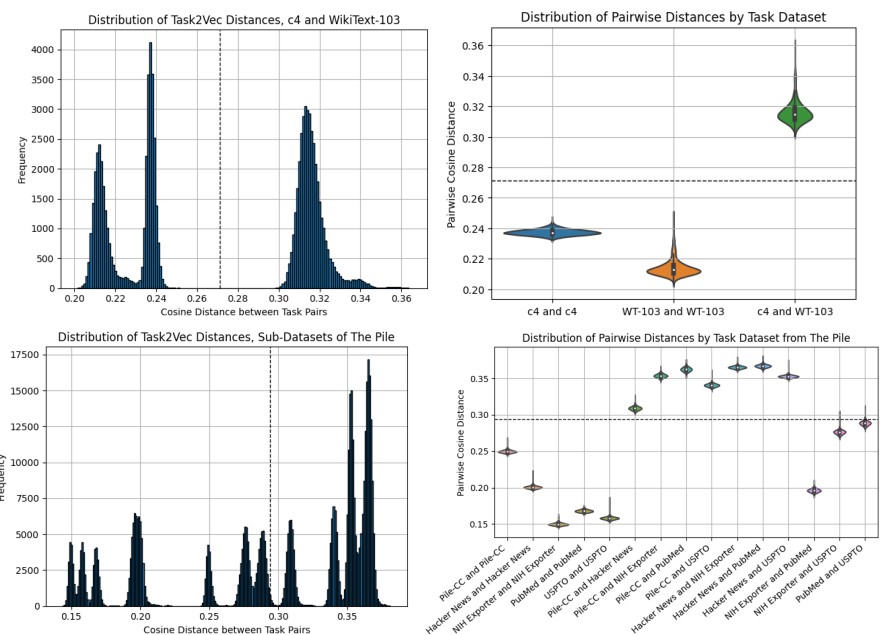

Figure 1: **Distribution of pairwise batch distances reflect conceptual and semantic dataset properties, therefore increasing trust in the diversity and cross diversity coefficient.** Pairwise task distances from concatenated C4 and WikiText-103 dataset (top) and concatenated five sub-datasets of The Pile (bottom) take on a multi-modal form according to dataset comparisons. Pairwise distances are segmented by source datasets for each pair of batches (right), where each sub-distribution corresponds to a mode from the histograms (left). Dotted lines denote the cross diversity coefficient of the concatenated C4 and WikiText-103 dataset (top) and concatenation of five sub-datasets of The Pile (bottom). These results show that combining batches from two different datasets computes a higher cross diversity, as expected. Therefore, these results align with human intuition, increasing the confidence in the diversity and cross diversity coefficients as a diversity metric.

**Results:** Our key observations are:

- The cross diversity coefficient for the C4 and WikiText-103 concatenated dataset is 0.2711, about +0.03-0.05 higher than that of each individual dataset.

- The cross diversity coefficient for the concatenation of the five sub-datasets of the Pile is 0.2939 (Table 1), which is about +0.04-0.1 (Figure 1) that of each individual dataset.

This increase in diversity occurs because concatenating datasets produces higher pairwise Task2Vec distances between batches from different datasets (see Figure 1). Note that, this aligns with human intuition that combining data from heterogeneous sources increases the overall diversity of the data.

### 3.3 DISTRIBUTION OF PAIRWISE BATCH DISTANCES REFLECTS CONCEPTUAL AND SEMANTIC DATASET INFORMATION

To increase our confidence in the diversity and cross diversity coefficient as diversity metrics, we study distributions of the Task2Vec (cosine) distances used to compute the coefficient. We examine the alignment of the grouping of these distances with (human) conceptual and semantic understanding in Figure 1.

**Experiments:** We analyze Task2Vec (cosine) distances between batches from five sub-datasets of The Pile. We compare distances between batches of individual sub-datasets and distances across different sub-datasets.

**Results:** Our key observations are:

- Figure 1 (top, left) shows 3 modes. We confirm that the modes correspond to pairings of datasets in Figure 1 (top, right). For instance, the right-most mode, corresponding to distances with values higher than the cross diversity coefficient, consists of pairwise distances between C4 and WikiText-103 batches. This confirms intuitive properties we'd expect, i.e. we'd expect 3 modes given 2 datasets ($C_2^2 + 2 = 3$).

- Similarly to the preceding point, Figure 1 (bottom, left) shows 15 modes, which is exactly the number expected in enumerating all possible pairs of batches from 5 datasets.[2] Due to overlaps in distance values we only see 11 modes in the Figure 1 (bottom, right).

- We expect pairings of unrelated datasets to have higher cross diversity compared to pairings of related datasets. We observe this in Figure 1 (right). For the concatenated dataset of C4 and WikiText-103, the distribution of pairwise distances where one batch is from C4 and one is from WikiText-103 (right-most violin) is higher than that of individual datasets. For the concatenated sub-datasets of The Pile, the violin plots for combinations of conceptually unrelated datasets group above the dotted line (e.g. Hacker News and PubMed), while the violin plots of technical subjects written similarly [3] are below the dotted line (e.g. PubMed and USPTO). Note however that all combined cross diversities always increased after a concatenation.

- We expect Pile-CC and HackerNews to cover the most diverse topics since they are broad web-scale datasets, unlike the remaining which are technical in nature. Therefore, we anticipate 1) these two to have the highest individual diversities, as shown in the first two violin plots in Figure 1, and 2) to have the highest increase when concatenated with other datasets, as shown in the 6th to the 12th violin plots when counting from the left, in Figure 1.

- Distances between batches from Pile-CC and HackerNews (sixth violin from the left) are the lowest among pairwise distances of concatenated datasets above the cross diversity coefficient. This aligns with human conceptual intuition because the Pile-CC and HackerNews are the most similar in those sub-datasets, since they are both web-scale datasets.

These findings build trust in the cross diversity coefficient as a dataset diversity metric, since the coefficient and underlying Task2Vec distances of batches behave in interpretable ways that align with human intuition. Since the diversity coefficient uses the same computational backbone as cross diversity, these findings also build trust in the diversity coefficient.

### 3.4 Diversity Coefficient Captures LLM Pre-training Data Distributional Properties

To instill further confidence in the diversity coefficient, we perform a correlation analysis with data distributional properties on the GINC dataset synthetic language dataset Xie et al. (2021). GINC generates sequences by modeling how real documents are generated given a fixed number of latent document concepts through a mixture of Hidden Markov Models (HMM), where each HHM has a latent concept that models document statistics, e.g. wiki bio. Further details on GINC can be found in section K.

**Experiments:** Given that each GINC dataset is a mixture of HMMs with a fixed number of latent concepts (1-10,000), we plot how the diversity coefficient varies as the number of latent concepts increases for each dataset. We plot this in Figure 2 (top) and fit a curve for GINC datasets with fixed vocabulary sizes of 50 and 150. Then we fix the number of latent concepts at 5 and 5000 and similarly plot how increasing the vocabulary size for the GINC dataset (50-10,000 unique tokens) increases the diversity coefficient. We plot this in Figure 2 (bottom) and fit a curve for GINC datasets with 5 latent concepts and 5000 latent concepts.

**Results:** Our observations are as follows:

- **Diversity coefficient increases with greater number of latent concepts.** Figure 2 (top) shows adding more latent concepts increases the diversity coefficient with diminishing returns. We hypothesize that additional latent concepts introduce new and varied document-level statistics, resulting in an increase in the diversity coefficient. The $R^2$ is high with values 0.952 and 0.898.

---

[2]Given a 5 by 5 distance matrix, we'd expect the lower triangular portion plus the diagonal to be the number of pairings, so $C_2^5 + 5 = 15$.

[3]e.g. NIH ExPorter and PubMed Abstracts both contain academic, medicine-focused writing, and have the lowest distances (third violin from the right) among combinations of different datasets.

- The diversity coefficient eventually saturates as more latent concepts are added. We hypothesize this is due to the small size of a synthetic data set vs. a real one.

- **Diversity coefficient increases with larger vocabularies.** Figure 2 (bottom) shows the measured diversity coefficient increases at a seemingly exponential pace for larger vocab sizes. The $R^2$ is high with values 0.993 and 0.984.

- We hypothesize the growth might be exponential because scaling the number of tokens produces a more diverse dataset by vastly increasing the number of ways to represent any sequence. More formally, given a sequence $x$ of length $T_x$ and vocab size $|V|$, the number of ways to represent that sequence is approximately $|V|^{T_x}$. Therefore, as $|V|$ increases, the growth rate of the exponential increases.

These results show the diversity coefficient successfully captures different distributional sources of variation of the data.

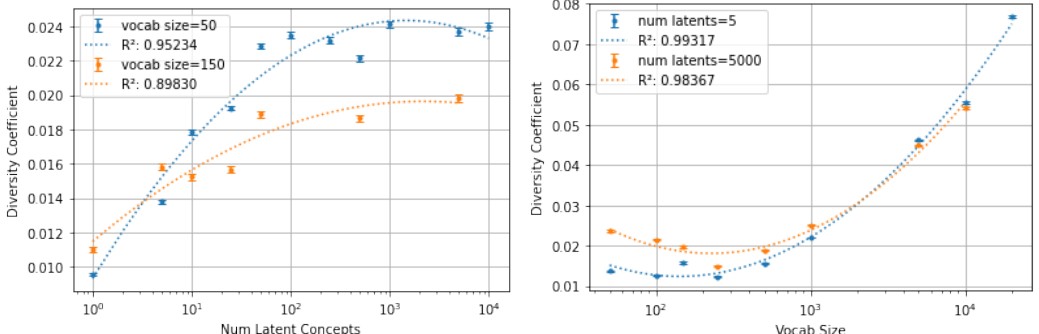

Figure 2: **Diversity coefficient of GINC datasets with varying number of latent concepts and vocab sizes shows the diversity coefficient behaves as expected.** The diversity coefficient increases and saturates with an increasing number of latent concepts (top) and exponentially increases with increasing vocab size (bottom). This implies that increases in the measured diversity coefficient correspond to changes in LM pre-training data distributional properties that intuitively enable more diverse data.

## 4 USING THE DIVERSITY COEFFICIENT IN PRACTICE: SETTING BATCH SIZE AND NETWORK PARAMETERS

**Experiments:** We test the sensitivity of the computed diversity coefficient value to changes in batch size and probe network parameters in order to gauge how these parameters should be set in practice for natural language datasets.

We vary the batch size and observe the impact on the diversity coefficient. For the same number of batches (200) and probe network (pretrained, fine-tuned GPT-2), we computed the diversity coefficient of C4 for batch sizes of 128, 256, 512, and 1024, and plot the results in Figure 3 (left).

We test the following probe network configurations to measure the diversity coefficient of C4 and of WikiText-103: 1. Pretrained GPT-2 with fine-tuning, 2. Pretrained GPT-2 without fine-tuning, 3. Randomly initialized GPT-2 with fine-tuning, 4. Randomly initialized GPT-2 without fine-tuning. Since using a random and/or non fine-tuned network is more resource efficient and easily accessible in practice, our motivation is to assess the necessity of using pre-trained and fine-tuned probe network, which is the original configuration used for Task2Vec in Achille et al. (2019). We aim to determine if a good approximation of diversity can be computed without fine-tuning. This setting is shown in Figure 3 (right).

**Results:** We observe that

- **Diversity coefficient increases with task batch size, but with diminishing returns.** Figure 3 (left) shows positive correlation between the diversity coefficient and batch size. This may be because larger batch sizes enable more unique tokens per batch.

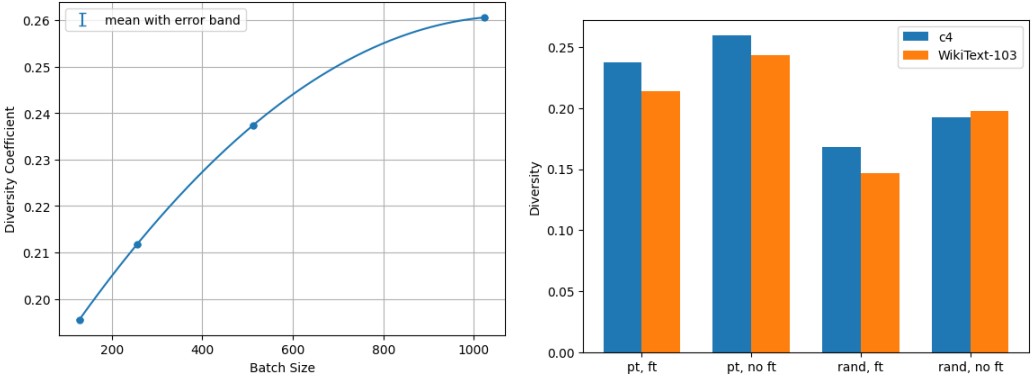

Figure 3: **Diversity coefficients of C4 computed using different task batch sizes show positive and diminishing returns with increasing batch size (left). Diversity coefficients of C4 and WikiText-103 computed using different GPT-2 probe network configurations show that random networks' estimates likely diverge from those of pretrained networks (in particular, they appear to underestimate diversity), and non-finetuned networks overestimate diversity vs. finetuned networks (right).** 95% confidence intervals for diversity coefficients are plotted, but are so small that they do not show. "pt" refers to pretrained network and "rand" refers to randomly initialized network. "ft" refers to a network that was finetuned per task and "no ft" refers to no finetuning performed.

- **Coefficients from using random probe networks likely diverge from using pre-trained networks.** Since the Task2Vec method (Achille et al., 2019) uses a pretrained fine-tuned network, we consider the diversity computed using this configuration as a source of truth. Figure 3 (left) shows that using a random probe network underestimates diversity compared to pretrained networks, which is in accordance with results from Miranda et al. (2022b) on vision datasets and indicative of random networks' estimates diverging from pre-trained networks'.

- **Using a non-fine-tuned pre-trained network overestimates diversity.**

- Trends in diversity coefficient overestimation vs. underestimation for different probe network configurations are consistent across C4 and WikiText-103.

Based on these findings, we recommend using a batch size of 512 sequences for faster computations and fewer out of memory issues and pre-trained fine-tuned network. Since the other setting's diversity difference is large from the ground truth Achille et al. (2019), we can't recommend it. If the intuitive properties reproduce for the other two options, we'd recommend it, but this is left for future work.

## 5 RELATED WORK

Existing diversity metrics have concentrated on data produced by Generative Adversarial Networks (GANs) and involve variations of a precision- and recall-based framework originally proposed in (Sajjadi et al., 2018) to measure quality and diversity, respectively (Kynkäänniemi et al., 2019; Simon et al., 2019; Naeem et al., 2020). Similar to our metric, these methods use embedding functions. These methods argue data quality is not synonymous with data diversity in the context of GANs (Fowl et al., 2020) and take a two-metric approach. Regarding LLMs, we argue that data diversity is a subset of data quality, which is demonstrably important to enable emergent capabilities such as in-context learning. Recent work has also confirmed the importance of diversity from the perspective of deduplication (Tirumala et al., 2023), and the general importance of quantitatively informed data selection (Xie et al., 2023). Hence, diversity metrics capture an important aspect of data quality.

A recently proposed diversity metric that doesn't rely on an embedding function is the Vendi Score (Friedman & Dieng, 2022), which is the exponential of the Shannon entropy of the eigenvalues of a similarity matrix/kernel. However, the benefits of this more sophisticated aggregation method are not clear, and its computation ($O(n^3)$) is more expensive than the diversity coefficient ($O(n^2)$). Moreover, it assumes a suitable similarity function/kernel, and does not provide guidance on data representation, arguably the most important ingredient in machine learning. Furthermore, they suggest that utilizing

data representational methods such as embedding networks that require pretrained models may be limiting. We argue instead that data representation is a fundamental property of data processing that has led to the overwhelming success in machine learning due to deep learning, e.g. in computer vision (Krizhevsky et al., 2012; He et al., 2015), natural language processing (Devlin et al., 2018; Brown et al., 2020; Chowdhery et al., 2022; OpenAI, 2023; Google, 2023), game playing (Silver et al., 2016; Mnih et al., 2013; Ye et al., 2021), theorem proving (Rabe et al.; Polu & Sutskever, 2020; Han et al.), code (Chen et al.) and more. Building on the success of deep learning data representations, we demonstrate deep learning is a strong way to create dataset/task embeddings. In contrast to the Vendi Score, our approach learns effective embeddings of datasets in an end-to-end manner, whereas the Vendi Score is focused on measuring diversity between specific data points. Since many datasets are publicly available (e.g. Common Crawl, Wikipedia), data used to train new models may be curated from such datasets, necessitating a metric that captures overall data diversity. These scenarios are thus in favor of using the Task2Vec diversity coefficient. Therefore, our method is likely more general and scalable than the Vendi Score. We leave a detailed comparison with the Vendi Score as future work.

## 6 DISCUSSION

Our work extends, examines, and thus validates the application of the Task2Vec diversity coefficient to a new modality – natural language – and demonstrates that open LLMs are pre-trained on formally diverse data. Through an extensive set of experiments that verifies intuitive properties of a diversity metric, we instill confidence in the diversity coefficient method, and therefore effectively formalize the concept of data diversity. Our conceptually well-motivated lower and upper bounds on the diversity coefficient aid in the understanding of the magnitude of the diversity coefficient. Although the bounds we propose only apply to sequence data with a symbolic vocabulary, using a multi-modal embedding method (e.g. a multi-modal probe network) can easily address this limitation.

One potential limitation of our method is the need for a data representation. Although the requirement for a data representation might seem restrictive, we argue that it is an inherent aspect of data processing. Choosing symbols (e.g., one-hot) or raw pixels (or anything else) **is** a choice of data representation. We suggest deep learning representations due to their overwhelming success in machine learning, e.g. in computer vision (Krizhevsky et al., 2012; He et al., 2015), natural language processing (Devlin et al., 2018; Brown et al., 2020; Chowdhery et al., 2022; OpenAI, 2023; Google, 2023), game playing (Silver et al., 2016; Mnih et al., 2013; Ye et al., 2021), theorem proving (Rabe et al.; Polu & Sutskever, 2020; Han et al.), code (Chen et al.) and more. In addition, widely available open-source pre-trained models (e.g. CLIP (Radford et al., 2021), LLaMA (Touvron et al., 2023), etc.) have made choosing a good embedding method easier. Another potential limitation of the diversity coefficient is the need to fine-tune a probe-network. Though this does introduce some computational overhead, it is relatively small (fine-tuning only the final layer) and outweighed by the utility of using information-rich Task2Vec embeddings.

Importantly, the relationship between pre-training dataset diversity and test performance remains a key question. To investigate, we conduct preliminary experiments, pre-training three GPT-2 models on datasets of varying diversities, then evaluate their performance on diverse validation datasets. We observe in Table 2 that increased diversity leads to decreased cross-entropy loss, indicating a positive relationship between diversity and model performance on diverse tasks. Although the cross-entropy values are arguably large, Figure 7 shows that models were still pre-trained enough to grasp important aspects of sentence syntax and structure of their respective dataset, indicating model performance on evaluation represents meaningful improvements in ability. Hence, we conjecture pre-training on higher diversity data improves test performance on diverse tasks, though more extensive experiments are needed to know so conclusively.

Data has taken a central role in the success of modern machine learning methods, like GPT4 OpenAI (2023), CLIP Radford et al. (2021), and PaLM 2 Google (2023), with special relevance for architectures with few inductive biases, like the popular Transformer (Vaswani et al., 2017). Therefore, it is paramount to understand the pretraining data we use, beyond scale alone. We conclude the diversity coefficient is a trustworthy metric, and conjecture the diversity coefficient can be used to build quality, diverse datasets for LLMs. We hope our contributions inspire more effective data collection and curation in machine learning that goes beyond scale alone to improve performance.

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
