# A  INTUITION OF TASK2VEC

To better understand the Task2Vec embedding, observe that the (diagonal) of the FIM can be interpreted as a measure of the information that a given parameter contains about the generative distribution $p_w(\hat{x}_t \mid x_{t-1:1})$. Therefore, it serves as a unique fingerprint, or feature vector, for a batch, which defines a task distribution; as such, this vector, in its entirety, is as large as the number of parameters of the probe network. Empirical findings in (Achille et al., 2019) show that Task2Vec embeddings cluster in a way that reflects semantics between different visual concepts, and that Task2Vec cosine distances are positively correlated with taxonomical distances.

We will provide a rephrasing from the original Task2Vec Achille et al. (2019) explaining why Task2Vec is captures the importance of the weights and therefore why it serves as a unique fingerprint for a task. The importance of a weight in a network can be measured by how much the predictions changes (average KL divergence of the original vs perturbed output). Consider a perturbation of the weights $w' = w + \delta_w$. This is the a 2nd-order approximation to the change in outputs:

$$E_{x \sim \hat{p}}[KL(p'_w || p_w)] = \delta_w F \delta_w^\top + o(delta(w)^2)$$

Therefore, the information content of the weights is measured by this change in output captured by the Fisher Information Matrix. Therefore, this serves as a unique finger-print of which parameters are important, making the diagonal of F a strong candidate for the unique fingerprint of a task via importance of weight.

Furthermore, we have found from initial experimentation that other straightforward embedding methods, such as using model activations, do not serve as reliable diversity metrics since, for example, they do not capture variation in data characteristics that align with human intuition.

# B  LLM PRE-TRAINING DATASETS

Since LLMs are often trained on internal, non-public datasets[4], we used publicly available language datasets from the same sources as LLM pre-training data:

**C4**, a 305GB cleaned version of Common Crawl's web crawl corpus in English Raffel et al. (2019). Sequences in C4 were extracted from the web via de-duplication methods and heuristics to remove boiler-plate and gibberish.

**WikiText-103**, a 500MB collection of over 100 million tokens extracted from the set of verified Good and Featured articles on Wikipedia Merity et al. (2016).

**The Pile**, a 825 GiB open-source English-text corpus for language modeling that combines 22 smaller, high-quality datasets from diverse sources Gao et al. (2020). These sources include Pile-CC (Common Crawl), PubMed Abstracts, Books3, OpenWebText2, ArXiv, and GitHub.

**SlimPajama**, an open-source reproduction of the LLaMA v1 dataset with extensively deduplication. It incorporates data from CommonCrawl (CC), C4, GitHub, Books, ArXiv, Wikipedia, and StackExchange, totaling 627 billion tokens Soboleva et al. (2023).

For instance, GPT-3 was trained on a filtered Common Crawl dataset and Wikipedia Brown et al. (2020), which are represented by C4 and WikiText-103. It was also trained on WebText2 and Books, which are sub-datasets of The Pile.

We also evaluate the diversity coefficient of the following six sub-datasets of The Pile:

**Pile-CC**, a 227 GiB preprocessed version of Common Crawl's web crawl corpus Gao et al. (2020). While both Pile-CC and C4 are sourced from Common Crawl, Pile-CC was preprocessed from Web Archive files, which are raw HTTP responses and page HTML, whereas C4 was preprocessed from WET files, which consist of plaintext. Nonetheless, we expect that both datasets are non-mutually-exclusive.

**HackerNews**, a 4 GiB scraped and parsed dataset of comment trees from Hacker News, a social news website that aggregates article links Gao et al. (2020). Articles are generally focused on topics in computer science and entrepreneurship.

---

[4]For instance, Gopher was trained on Google's internal dataset MassiveText.

**NIH ExPorter**, a 1.9 GiB dataset of NIH Grant abstracts for awarded applications from 1985-present hosted on the ExPORTER initiative Gao et al. (2020).

**PubMed Abstracts**, a 19 GiB dataset of abstracts from 30 million publications in PubMed Gao et al. (2020).

**USPTO Backgrounds**, a 23 GiB dataset of background sections from patents granted by the United States Patent and Trademark Office (USPTO) Gao et al. (2020).

**OpenWebText2**, a 38 GiB dataset based on data extracted from Reddit posts, deduplicated, and filtered for English content using FastText. Strict filtering with local-sensitive hashing (LSH) was done and only unique content with similarity of less than 0.5 was used. The finalized dataset comprises 38 GiB.

## C    FUTURE WORK

Our future research will explore the potential of the Task2Vec distance function for pre-training dataset curation. Given that the objective of pre-training is to maximize downstream task performance, we define high-quality training data as data that facilitates the best achievable performance on such tasks. We anticipate that higher diversity in the dataset will increase the likelihood of achieving this objective. The rationale is that a higher data diversity implies a broader coverage of tasks or batches, thereby increasing the probability of training the model on tasks or data representations that are relevant to evaluation tasks. Our focus will be to leverage Task2Vec to assess the similarity between individual data points, batches, or datasets to a target task. This assessment will enable us to curate the training data by selectively removing tasks that resemble random, noisy, or irrelevant sequences, which may adversely affect downstream performance.

## D    FURTHER JUSTIFICATION AND EXPLANATION OF CROSS DIVERSITY

We include the notion of the cross diversity coefficient as it also leverages the ability of distance between Task2Vec embeddings to capture important properties of the data, and allows one to more clearly assess similarity/diversity *between* datasets (hence the term *cross* diversity), as opposed to similarity/diversity *within* datasets (the focus of the "normal" diversity coefficient). Thus, when information on clearly defined sub-datasets is available, one can use cross-diversity to more specifically determine the diversity of the concatenation of the given datasets (to form the total/overall dataset). Thus, the cross diversity coefficient provides us the opportunity to more robustly characterize methods of defining data diversity by offering a different perspective on combined datasets from the diversity coefficient. Note that in the case $D_1 = D_2$, i.e. one is calculating the cross diversity of a single dataset (with itself), cross diversity becomes equivalent to diversity.

## E    TASK2VEC DIVERSITY COEFFICIENT CORRELATES WITH GROUND TRUTH DIVERSITY

As shown in Miranda et al. (2022b), when the ground truth diversity is available for a synthetic Gaussian benchmark, the Task2Vec diversity coefficient correlates with the ground truth diversity. These results provide confidence in the Task2Vec diversity coefficient as diversity metric.

## F    PIPELINE FOR DIVERSITY COEFFICIENT COMPUTATION OF NATURAL LANGUAGE DATASETS

Figure 5 shows our pipeline for computing the diversity coefficient of large scale, natural language datasets. See section 2.2 for more details on our method.

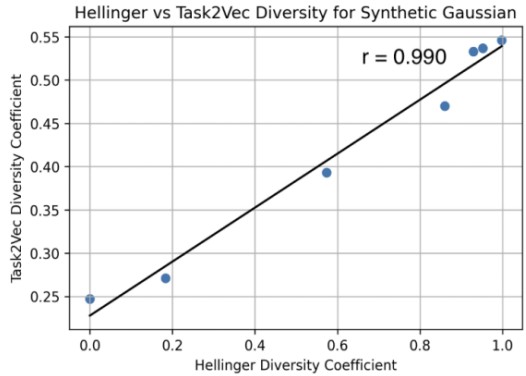

Figure 4: **Task2Vec diversity coefficient correlates with ground truth diversity for synthetic Gaussian benchmark.** Source: Miranda et al. (2022b)

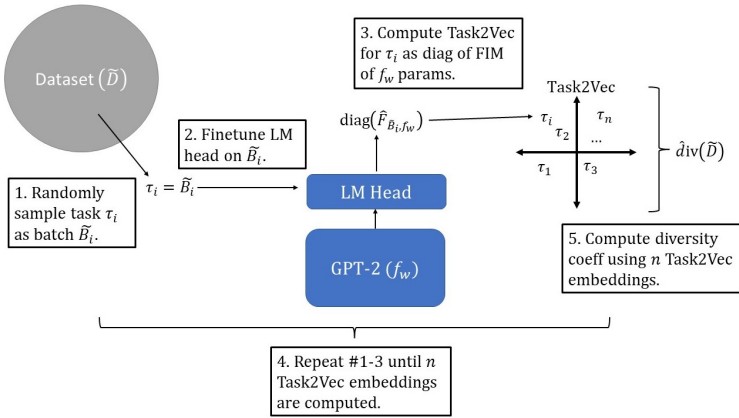

Figure 5: **A depiction of a pipeline to compute the Task2Vec diversity coefficient for a natural language dataset.**

## G    ROLE OF THE DIVERSITY OF PRE-TRAINING DATA ON TEST PERFORMANCE

Table 2 demonstrates preliminary experiments and results on the relationship between pre-training data diversity and downstream evaluation performance, particularly evaluating on tasks from diverse sources.

| Training Data Set | Pile-CC (**div 0.250**) | OpenWebText2 (**div 0.222**) |
|---|---|---|
| USPTO (**div 0.158**) | 5.9988 | 6.4414 |
| PubMed (**div 0.168**) | 6.1412 | 6.4204 |
| USPTO + PubMed (**div 0.174**) | **5.7954** | **6.1815** |

Table 2: **The table illustrates that as the diversity coefficient of training data increases, the cross-entropy loss (CE) decreases on diverse tasks, implying that training on higher diversity data improves evaluation performance on diverse tasks.** Each model is trained for the same number of tokens (1.31 B tokens total), all models have 51.5 M parameters, and all use a GPT-2-based architecture; all other hyperparameters are identical and controlled for. We evaluate on Pile-CC (Pile Common Crawl) and OpenWebText2 since those datasets align with intuitively diverse datasets, which we verify with the computed diversity values shown.

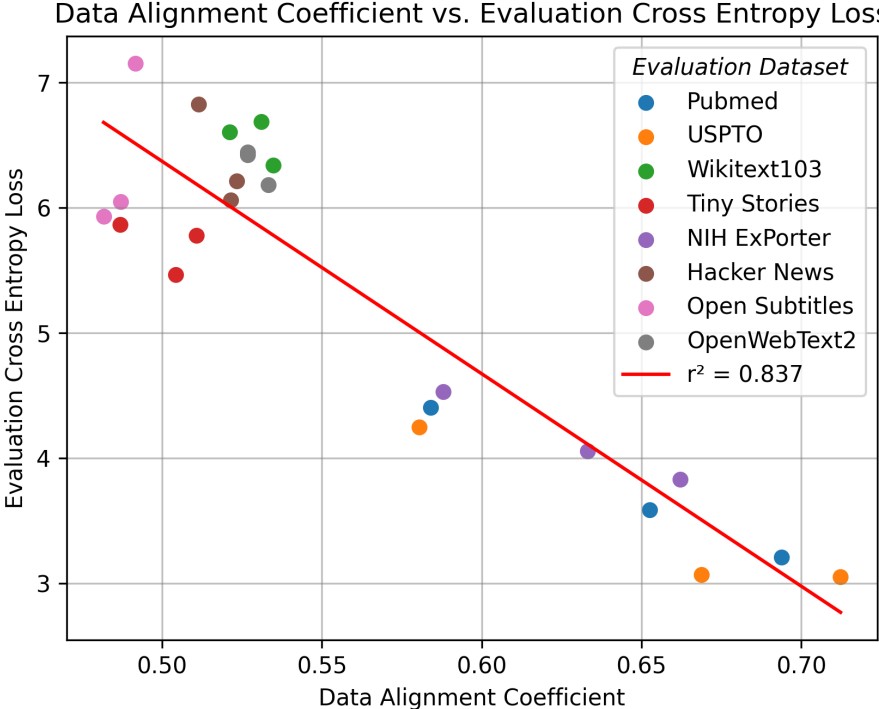

Figure 6: **The data alignment coefficient demonstrates a strong relationship with model performance (cross entropy loss) on various evaluation datasets** ($r^2 = 0.837$). The data alignment coefficient is computed between a model's pre-training dataset (PubMed Abs., USPTO, or PubMed Abs. + USPTO) and a single evaluation dataset (represented by a unique color).

## H    ROLE OF DATA ALIGNMENT COEFFICIENT ON TEST PERFORMANCE

Given a distance function for a batch of data (or task) one can also measure how aligned or similar two data sets. One can do this by computing the cross diversity between batches/tasks of data and then subtract one from it. Therefore, we propose one minus the cross diversity coefficient as the definition of the alignment coefficient:

$$\hat{align}(D_1, D_2) = 1 - \mathbb{E}_{B_1 \sim D_1, B_2 \sim D_2} d(\vec{f}_{B_1}, \vec{f}_{B_2})$$

We choose this method since:

1. It is simple to define given the definition of cross diversity proposed in this paper.

2. It provides simple confidence interval to compute given that it's an expectation – in contrast to using a large batch of data and the cosine distance of two Task2vec batch/task embeddings.

Figure 6 demonstrates that there is a moderate-strong relationship between the alignment coefficient (between pre-train data and evaluation data) and model performance (cross entropy loss) on various evaluation datasets ($r^2 = 0.80, r = -0.89$). The raw alignment values with 95% confidence are reported at H and, as expected, when datasets share similarities in topic and structure, the alignment coefficient is higher (see caption of table). In addition, Figure 7 shows that despite the arguably high loss, the models are pre-trained enough to grasp important aspects of sentence syntax and structure of their respective dataset.

| USPTO Evaluation Example | Wikitext-103 Evaluation Example |
|---|---|
| **Context**:
`1. Field of the Invention\nIn an extensive plant breeding program, Grant Merrill, originator and now deceased, originated a large number of new and distinct varieties of fruit trees, and which included the herein-claimed variety of peach tree. Such plant breeding program was undertaken in originator's experimental orchard` | **Context**:
`Traditional Chinese literary criticism emphasized the life of the author when interpreting a work , a practice which Burton Watson attributes to " the close links that traditional Chinese thought posits between art and morality " . Since many of Du Fu 's poems feature morality and history , this practice is particularly important . Another` |
| **Pubmed-trained model output**:
`, and the results of the present study suggest that` | **Pubmed-trained model output**:
`aspect of the author's work is that the author` |
| **USPTO-trained model output**:
`plant breeding program.\n2. Description of the` | **USPTO-trained model output**:
`problem with the present invention is that the present invention` |
| **Pubmed & USPTO interleaved-trained model output**:
`breeding program.\n2. Description of the Related` | **Pubmed & USPTO interleaved-trained model output**:
`issue is the fact that the author is not familiar` |
| **Ground truth text**:
`located near Exeter, Tulare County, Calif.\n2. Prior` | **Ground truth text**:
`reason , identified by the Chinese historian William Hung , is` |

Figure 7: **Despite smaller scale and relatively high loss/perplexity on certain evaluation datasets, the experimental models still grasp important aspects of sentence syntax and structure, even on evaluation data they otherwise struggle on (e.g. Wikitext-103, for which all models scored around 6.6 loss or more)**. In addition, when evaluated on similar, well aligned datasets (e.g. USPTO + PubMed Abs. (train) on USPTO (validation)), models are sometimes also able to match the form and semantic content of the example. Input examples were randomly picked from the given evaluation dataset.

| Pre-training dataset | USPTO (validation) | PubMed Abs. (validation) | Openwebtext2 | NIH Exporter | Hacker News | Open Subtitles | Wikitext-103 | Tiny Stories |
|---|---|---|---|---|---|---|---|---|
| USPTO | $0.7123 \pm 0.001717$ | $0.5840 \pm 0.001389$ | $0.5267 \pm 0.001377$ | $0.5879 \pm 0.001388$ | $0.5234 \pm 0.001275$ | $0.4917 \pm 0.001162$ | $0.5311 \pm 0.001303$ | $0.5107 \pm 0.001203$ |
| PubMed Abs. | $0.5805 \pm 0.001396$ | $0.6939 \pm 0.001697$ | $0.5268 \pm 0.001367$ | $0.6622 \pm 0.001569$ | $0.5114 \pm 0.001300$ | $0.4817 \pm 0.001145$ | $0.5212 \pm 0.001200$ | $0.4868 \pm 0.001167$ |
| USPTO + PubMed Abs. | $0.6687 \pm 0.001602$ | $0.6526 \pm 0.001513$ | $0.5332 \pm 0.001390$ | $0.6331 \pm 0.001452$ | $0.5215 \pm 0.001272$ | $0.4871 \pm 0.001123$ | $0.5347 \pm 0.001290$ | $0.5042 \pm 0.001169$ |

Table 3: **The data alignment coefficient appears to capture an intuitive notion of data similarity, since it finds training data that shares similar semantics and structure as the validation data as most aligned.** In particular, PubMed Abs. (train) and NIH Exporter, which share the semantics of health-related research and the structure of being research writing, are found to be more aligned than USPTO (patent application backgrounds). Similarly, USPTO + PubMed Abs. (train) is more aligned to USPTO (validation) than PubMed Abs. (train), but less aligned to USPTO (validation) than USPTO (train), as expected. Each cell indicates the alignment coefficient between the given pre-training dataset (row label) and evaluation dataset (column label).

# I    EXPERIMENTAL DETAILS

## I.1    DATASET PREPROCESSING

In accordance with Achille et al. (2019), we used the training split of datasets to finetune the probe network when computing Task2Vec embeddings per dataset. Sequences were tokenized using a pre-trained HuggingFace GPT-2 tokenizer based on byte-level Byte-Pair-Encoding, and padded or truncated to a max length of 128. Because the WikiText-103 dataset contained empty text examples, we removed these examples before sampling batches to compute embeddings.

## I.2    MODEL ARCHITECTURE AND FINETUNING

We used a pre-trained GPT-2 model with a language modeling (LM) head on top. The pre-trained GPT-2 model itself has 12 layers, 12 heads, 768-d hidden size, and 117M total parameters. The LM head is a linear layer with weights corresponding to the input embedding layers. The model was pre-trained on the English language and the pre-trained GPT-2 tokenizer has a vocab size of $\approx 50k$ tokens. For all finetuning experiments, we fine-tuned only the LM head for 10 epochs. We used no learning rate scheduler and no gradient accumulation. We used the AdamW optimizer, since AdamW has been shown empirically to give better training loss and improved generalization.

We note that, in principle, the Task2vec diversity coefficient can be computed with any LLM. The metric itself is not specific to any particular LLM architecture or model version. We chose GPT-2 for our experiments due to computational efficiency and resource constraints. However, more powerful LLMs like LLaMA can also be used to compute the diversity coefficient. As long as the probe network used is consistent across experiments, the relative differences in the diversity coefficient value between datasets are directly comparable. The same goes for using pretrained vs. non-pretrained probe networks.

## I.3    NUMBER OF BATCHES AND BATCH SIZE SELECTION

Diversity coefficients in Table 1 were computed using randomly selected batches of size 512 sequences and a pre-trained, finetuned GPT-2 probe network. Diversity coefficients of C4, WikiText-103, The Pile, Pile-CC, HackerNews, NIH ExPorter, PubMed Abstracts, and USPTO were each computed using 200 sampled batches. Given resource constraints, we found 200 batches[5] to be a sufficiently large number of batches to estimate the diversity coefficient with tight 95% confidence intervals on the order of 1e-5. We chose 512 as the batch size, since it is a relatively large and feasible batch size to fine-tune the probe network on 200 batches using Azure NV12s_v3 instances equipped with Tesla M60 GPUs in a reasonable amount of time (30+ hours).

## I.4    DIVERSITY COEFFICIENT COMPUTATION OF CONCATENATED DATASETS

The diversity coefficient of a concatenated dataset of C4 and WikiText-103 was measured over a combined set of batches. Each batch consisted of sequences sampled from one of these datasets, e.g. a batch could have sequences randomly sampled from C4 or WikiText-103 but not both. The coefficient was computed over 400 batches of batch size 512 ($20\overline{0}$ batches from each dataset). Note that for the concatenated dataset, we utilized the same 200 batches per dataset that were used to compute the coefficients of C4 and of WikiText-103 individually.

The diversity coefficient of concatenated five sub-datasets of The Pile was computed over 1000 batches (200 batches from each dataset) of batch size 512. Similarly to the concatenated dataset of C4 and WikiText-103, we utilized the same 200 batches per dataset that were used to compute the coefficients of each individual sub-dataset.

## I.5    DIVERSITY COEFFICIENT OF THE PILE VS. CONCATENATION OF FIVE SUB-DATASETS

We make a clarification on the approach taken to evaluate the diversity coefficient for The Pile vs. for concatenation of its five sub-datasets.

---

[5]This results in $(200^2 - 200)/2 = 19{,}900$ pairwise distances used to compute the diversity coefficient.

The diversity coefficient of The Pile was computed over 200 batches sampled across all 22 sub-datasets of The Pile. This means that any given batch could contain sequences across all 22 sub-datasets, i.e. a batch could have sequences from Pile-CC, HackerNews, and NIH ExPorter.

The diversity coefficient of the concatenated dataset was computed over 1000 batches comprised of 200 batches separately sampled from each of the five sub-datasets. Each batch contained sequences from only one sub-dataset, i.e. a batch could only have sequences from Pile-CC or HackerNews or NIH ExPorter.

We hypothesize this distinction in the diversity coefficient computation explains why the concatenated dataset has higher diversity, even though it consists of only five of the 22 sub-datasets of The Pile. For the diversity coefficient of The Pile, because batches were sampled such that any batch contains sequences from across the 22 sub-datasets, the batch representations learned by the probe network may have been more similar, resulting in lower diversity relative to the concatenated dataset.

## I.6 Details on Mixes used for Interleaved Datasets

"C4 and WikiText-103 (Mix1)" denotes an interleaved dataset, i.e. a dataset where examples from each sub-dataset are randomly mixed together according to their proportion in the mix, so that if you iterate through each example in a 1:1 mix then you might see e.g. Dataset1 example, Dataset2 example, Dataset1 example, Dataset1 example, Dataset2 example. Mix1 indicates that this interleaved dataset is 75% C4 examples and 25% WikiText-103 examples.

"Combination of five datasets (Mix2)" also denotes an interleaved dataset. Mix2 indicates that each sub-dataset constitutes X% of the interleaved dataset, where X% is chosen to as closely match LLaMAv1's data training mix as possible. For example, if LLaMAv1 is composed of twice as much web crawl data as Wikipedia data, then our Mix2 will have 2 Pile-CC examples per 1 WikiText-103 example.

## J Pairwise Distance Distributions of C4, WikiText-103, and The Pile

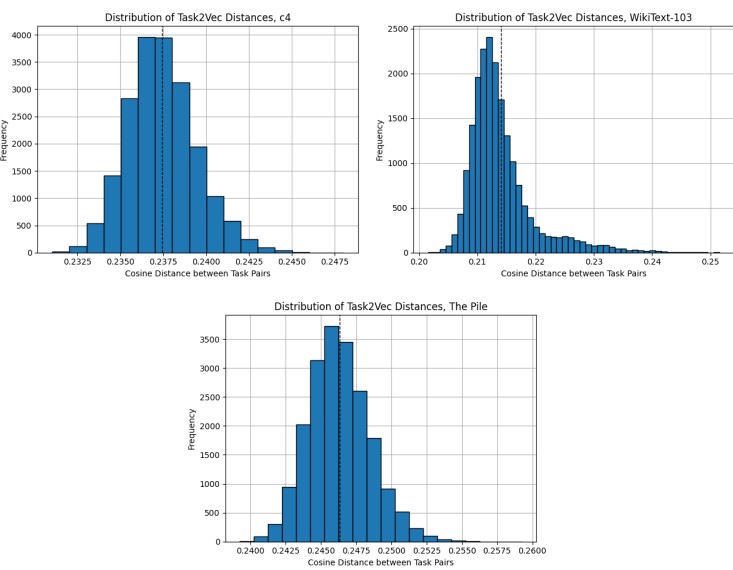

Figure 8: **Distributions of pairwise batch distances from C4 (top left), WikiText-103 (top right), and The Pile (bottom) are approximately Gaussian, which justifies the use of a sample of batches to measure the diversity coefficient.** Dotted lines indicate the average distance, i.e. the diversity coefficient, for each dataset.

**Experiments:** To provide confidence in the magnitude of the coefficient values of C4, WikiText-103, and The Pile, we plot the distribution of distances per dataset in Figure 8. We aim to show that a subsample of batches can provide a good estimation of population statistics, such as the diversity coefficient, which measures the expected Task2Vec (cosine) distance between batches.

**Results:** For each dataset, the pairwise distances take on unimodal and approximately Gaussian distributions with few outliers. These results suggest the Task2Vec distances are approximately normally distributed. This suggests we can make strong inferences about the population. Specifically, we are able to compute a good estimate of the diversity coefficient using 200 batches using the mean. This is in fact the same argument from Miranda et al. (2022a) – but we verified it applied in our setting. Figure 8 also shows few outlier batches – the presence of which could influence the computed diversity coefficient. This provides further confidence in the coefficient values computed and justifies our use of a sample of batches to estimate diversity.

**OpenWebtext:** Data from Reddit post URLs was extracted, deduplicated, and filtered for English content using FastText. Web pages were pulled using the newspaper python package, and near-duplicates were identified and removed using local-sensitivity hashing (LSH). Only documents with a unique content similarity of less than 0.5 and more than 128 tokens were retained. The finalized dataset comprises 38GB from 8,013,769 documents. Annotations: None present in the dataset. Used to train GPT2.

## K    GENERATIVE IN-CONTEXT LEARNING (GINC) DATASET

### K.1    BACKGROUND

The GINC dataset is generated using the latent concept framework proposed in Xie et al. (2021), where language models condition on a prompt to infer latent document concepts learned during pre-training. The pretraining distribution is defined using a uniform mixture of Hidden Markov Models (HMMs) parameterized over a family $\Theta$ of latent concepts.

### K.2    DEFINITIONS OF GINC DATASET PARAMETERS

**Number of latent concepts:** A latent concept $\theta$ parameterizes the transitions of a HMM in the mixture. A latent concept (e.g. a wiki bio) contains document statistics, such as semantics, syntax, and the formatting of and distribution of tokens.

**Vocabulary size:** Each HMM in a given mixture outputs a fixed number of tokens, defined as the vocabulary size. The vocabulary is generated by enumerating combinations of letters from a to z, aa to az, etc. The delimiter token is designated by a backslash. Sequences are tokenized by whitespace.

### K.3    SUPPLEMENTAL FIGURES FOR DIVERSITY COEFFICIENT VS. GINC PARAMETERS

Figure 9 confirms that the trends between the diversity coefficient and number of latent concepts (left) hold even as vocab size is varied. Similarly, trends between the diversity coefficient and the vocabulary size (right) hold as the number of latent concepts is varied. These trends were noted in Section 3.4.

## L    DISCUSSION (CONT.)

Our paper introduces a metric that leverages tunable parameters, such as the number of batches, batch size, probe network configuration (pre-trained vs. random, fine-tuned vs. not) and depth. While these elements influence the diversity coefficient's absolute value and necessitate the recalibration of lower and upper bounds (see Appendices I.3 and 4), a consistent choice of hyperparameters can mitigate these effects.

Intriguingly, our proposed diversity may not always correlate with model performance, as high diversity could simply be due to uniform noise. Nevertheless, we contend that a higher diversity, in the context of a sufficiently large model, likely indicates superior performance and data quality.

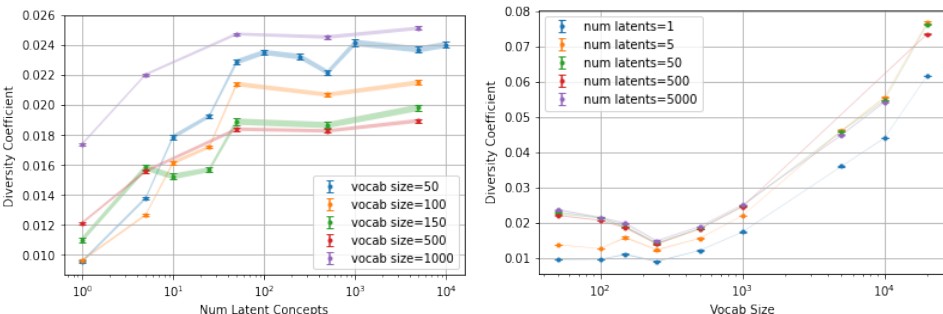

Figure 9: **Trends noted in Section 3.4 are consistent for diversity coefficient vs. number of latent concepts (left) and coefficient vs. vocab size (right) when the other parameter changes.** The diversity coefficient with 95% confidence intervals saturates with increasing number of latent concepts (left) even as vocab size is varied between 50-1000. Larger vocab sizes generally produce higher diversity coefficients (right) even as the number of latent concepts is varied between 1-5000.

Furthermore, our diversity metric is intentionally designed to be widely applicable, albeit concealing causal factors, rendering it an effective tool for ablation studies.

Despite our diversity metric's broader applicability, it may obscure certain causal factors. This limitation is intentional to enhance its practical usage – since causality is often difficult to infer and is out of scope. This can be overcome with data property ablation studies, as we showed in our GINC dataset experiments.

Currently, our proposed bounds are specific to sequence data with a symbolic vocabulary, limiting their applicability across different modalities. To overcome this limitation, we suggest using a multimodal embedding method for embedding diversity coefficients and lower/upper bounds across tasks.

To really clarify why FIM is better than activations, we provide this intuitive explanation. FIM gives a weight/feature of which parameter of the generative distribution matters, e.g. the first coordinate of Task2Vec corresponds to how artsy the text sequence is. This is a feature of a task or dataset itself. Therefore, FIM exactly approximates the (task) data generative distribution we are trying to embed. Therefore, we conjecture it results in superior representations for datasets compared to activations since it directly approximates the data (or task) generative distribution. Our study, and references, provide positive evidence in favor of this argument.

The strength of embeddings is their ability to approximate **semantics** in a way that symbols may struggle with, such as distinguishing the equivalence of two sentences with different symbols but identical meanings. In NLP there is no easy way to determine this equivalence. In formal mathematics, symbolic semantics and thus equivalence can sometimes be done exactly. Though it does not come without its costs, e.g. requires expert knowledge, computationally demanding or (approximately) exhaustive representations like e-graphs. Therefore, embedding methods for data diversity, quality, etc. have the unique advantage of being more generally applicable.

Our diversity calculations predominantly utilize a small model (GPT-2). Despite the ongoing discussion concerning the emergence of large language models (LLMs), our conjecture extends the results to models of all sizes. We base this inference on the fact that the manifestation of emergence is intimately tied to the specific metric employed, and the sudden unpredictable jumps disappear when smooth metrics are applied Schaeffer et al. (2023). The cosine distance is smooth and does not have this issue.

**Why and when does diversity matter?** We propose two central conjectures for the importance of diversity and provide the underlying rationale:

1. **Conjecture 1: Diversity is essential because it promotes learning-to-learn (a surrogate for General Intelligence).** The main argument is that a significant level of diversity corresponds to a multitude of tasks in the dataset. Therefore, to achieve high (test) performance, the model must perform well on all tasks. One potential strategy is by learning-to-learn,

thereby allowing transferability when tasked with a new problem. Another alternative could be to memorize all tasks.

2. **Conjecture 2: Diversity is crucial because it enhances the probability that the pre-training set covers the test set.** Diversity is a formal score of coverage – it aims to reflect the effective number of tasks in a dataset. Thus, increased diversity equates to more tasks in a dataset. This (could) boosts the chance of the training set covering the test set, hence improving performance, given a sufficiently large model like an LLM. The direct exploration of this conjecture is slated for future investigation, but we provide a suggestive (correlative) analysis of one reason why LLMs might perform so well.

Another benefit is that our method does not rely on activations from an arbitrarily selected layer in a network. Lastly, note that activations may be unreliable for embedding dataset/tasks because large distances between datasets/tasks may be due to well-separated decision boundaries instead of intrinsic semantic properties of the dataset/task. In contrast, the diversity coefficient is well-justified, extensively tested in our work and previous work, e.g. the diversity coefficient correlates with ground truth diversities, cluster according to semantics, taxonomy etc. (see section E and Achille et al. (2019); Miranda et al. (2022a)). In short, FIM-based representations are motivated by information theory (e.g. FIMs are metrics in distributions) and have been extensively tested by independent sources (Miranda et al., 2022a; Achille et al., 2019; Vu et al., 2020).

**Limitations:**

- The diversity coefficient presents an aggregate measure that masks the underlying causal factors. Despite this, we illustrate how it might be employed to uncover these factors. We show this through the use of vocabulary size and latent space, acknowledging that these experiments could be resource-intensive. Causality is a challenging topic, and we do not claim to solve it through our experiments. Our experiments in this regime are mostly to show that the diversity coefficient (might) correlates/captures different sources of diversity beyond number of concepts or tasks.

- The computation of Task2Vec embeddings requires more resources than computing simply the activations. However, given the proven correlation with ground truth task generative parameters from previous studies, we posit that it supersedes activations. Furthermore, we hypothesize that using activations could result in high distances due to optimization for decision boundaries, making it less reliable for measuring distances i.e., high distances in activation space might be artificially high. We observed this but plan to give more detailed study in the future.

- The choice of an expectation as the aggregation function could be seen as arbitrary. Alternatives such as the Vendi score are promising, but still under-explored and computationally demanding compared to expectations/sums. Future work could focus on the comparative analysis of the total distance sum and the Vendi score. We hypothesize, in line with the Central Limit Theorem (CLT), that the results might not differ significantly e.g., CLT still converges to (unit) Normal given the proper normalization procedure.

- We reject the notion that the use of models is a limitation. As discussed earlier, models can provide superior data embeddings, and all forms of data representations are always required. For example, the identity function or symbols are data representations.

**Implications:**

- Given the impressive performance of LLMs, our study suggests a correlation with our diversity measure, potentially providing an explanation for this high level of performance.

- High diversity implies a larger task coverage. Therefore, we conjecture that a highly diverse pre-training set could increase the probability of including relevant pre-training data for the target task/testing. This suggests that collecting more diverse data could be a method to enhance performance. If the model is sufficiently large, we conjecture this method always (stochastically) monotonically increases the performance (as implied by (Zhang et al., 2021)).

- The transition from a qualitative to a quantitative measure of diversity can be seen as a significant advancement in the field because of conceptual transitions about how we think and talk about data quality/diversity.

- The use of Task2Vec to embed data implies a method applicable to any modality, potentially benefiting all areas of machine learning research.