# OpenReview forum: "Beyond Scale: the Diversity Coefficient as a Data Quality Metric Demonstrates LLMs are Pre-trained on Formally Diverse Data"
_ICLR.cc/2024/Conference — Submitted to ICLR 2024_

### Official Review · Reviewer_Z6o3 · 2023-10-16

**Soundness:** 3 good
**Presentation:** 2 fair
**Contribution:** 3 good
**Rating:** 6
**Confidence:** 4

**Summary:**

This paper proposes a ‘diversity coefficient’ that can measure the diversity in certain dataset and a ‘cross diversity coefficient’ that measure the difference between two datasets. Experiments show that the coefficient has three ideal characteristics: the coefficient increases as more datasets are concatenated, the number of latent concepts increases, and a richer vocabulary is used. Besides, the proposed efficient also show that the current public pre-training datasets has relatively high diversity, which could explain why we can get good LLM trained on them.

**Strengths:**

1) The dataset quality is a very important topic but unfortunately not well-defined and studied. It is great that this paper devotes to this novel and critical topic.

2) Measuring the dataset quality in terms of diversity is a nice and neat idea, authors also propose ideal characteristics of the diversity metric and ways to justify them.

**Weaknesses:**

1) I understand that authors want to show the diversity coefficient can work by comparing real data diversity with theoretical lower and upper bound. However, if we treat lower bound as baseline, it is too weak to say the dataset having better diversity is really diverse?

2) Since this approach requires finetuning LM, it is actually introducing some computational overhead. I understand this overhead is not large, but it should be stated in the limitation section.

3) More recent data-centric related works should be added. Most related works listed are before 2022. Only two citations are after 2023, one is GPT-4, and the other is Palm 2, neither of which are references to Data Centric methods. To my knowledge, there is at least one paper about increasing data diversity from Meta AI in 2023.08. “D4: Improving LLM Pretraining via Document De-Duplication and Diversification”. Probably can add to the Related Work?

4) Since the coding data is increasing important, authors should provides more insights about the diversity of code pre-training dataset.

5) Some conclusions are not well-justified.  In appendix G, experiments conclude that ‘higher data diversity leads to higher test performance’, but fail to ablate the influence of dataset overlap. In fact, the Pile dataset itself contains both the USPTO and PubMed, so I think model trained on the ‘USPTO+PubMed’ is better than models trained on either of them cannot purely attribute to the diversity increase. It’s possible that the high diversity is not the key factor, the actual factor is high diversity increase the overlap between train and test set, and therefore increase the test performance. In addition, the table in Appendix G should be further clarified. It’s hard to get which dataset is the training dataset and which one is the eval set.

In general, I think this paper is good for me. Although there are some unclear statements, it is okay because this topic is very new and not well-defined. I can consider further increasing the score if the authors solve my concerns well.

**Questions:**

1) The introduction of Task2Vec in the ‘method’ section is brief and should mention the further explanation in the Appendix A(I got stuck in this part for a while). Besides, I am kind of confusing about the dimension of vector’s embedding. According to the formula I think the diagonal dimension should equal to the number of model’s parameters. If that’s true the vector will be extremely large…

2) The setting “Randomly initialized GPT-2 without fine-tuning” in Section 4 makes me confused. Does that mean you only reset the LM head or the whole model? My concern is different random initialization could lead to very different results. So it’s not safe to conclude that ‘using a random probe network (always)**underestimates** diversity’.

3) What’s the relationship between the diversity efficient and the cross diversity efficient? What insight can we get by comparing them together?

4) small typo, ‘due’ should be ‘due to’ in Sec 3.4

5) In table 1, why concatenating C4 and Wikitext-103 decrease the Div Coeff? Any analysis on this exception?

6) In table 1, what does the ‘Combination of Five Datasets(MIX2)’ mean? The explanation of MIX2 setting is not very clear to me. Why only five datasets? Why 0.77 vs 0.23?

---

> ### Author Response · Authors · 2023-11-20
> **Response to great questions**
>
> We thank Reviewer Z6o3 for their time and feedback! Replying to each comment, suggestion and/or question in turn:
>
> This is an interesting point! One of the main reasons we used the theoretical lower/upper bounds is because the diversity coefficient is a completely new metric for quantifying diversity formally. Therefore, before this work there was no sense what the minimum/maximum would be for this metric. They would look like random number without no reference even if we used any real data set. Therefore, we verified that 1. the metric indeed behaves well 2. and even though the lower/upper bounds are adversarial, given a real sense of if the diversity numbers the coefficient reports behave well. We do acknowledge that the lower bound might be a simple baseline but believe it’s a crucial baseline due to the novelty of the diversity coeff. However, we did try to provide a real world sense/baseline by providing 12 different real world NLP data data sets – where 3 (NIH, USPTO, PUBMED) are single topic/area data sets. Using this we can see the ratio of the least diverse to most diverse is d(Pile-cc)/d(NIH)=0.2497/0.15=1.67, which still suggests the web crawled type data are nearly twice as diverse! We are happy to emphasis this later point and include in the discussion section the nuances and importance of the upper/lower bounds in our work.
> Thank you for pointing it out! We will include this weakness in the discussion and include in the future work that we plan to study the effectiveness of our method without any fine-tuning and only with a zero shot prompt. Though despite using fine-tuning we only fine-tune the final layer!
> We are happy to compute diversities for pre-training data sets coding datasets, great idea!
> Great observation! We are happy to address it clearly in the revision. We tried addressing it via two ways: 1. we used OpenWebText2 to avoid overlaps with the training set and we still observed the described trend (with lower CE losses). 2. In figure 6 in the appendix, we computed the “alignment coefficient” of the train and the test data and observed a positive correlation of r=0.8, meaning that the story is more nuanced as you hinted & alignment in train and test are important. 3. we do want to emphasizes that we used the test sets at eval so although there is overlap in terms of concepts/type of data there isn’t “cheating” in that we didn’t train in the test set. In addition, when the data is more diverse the likelihood of training on more relevant data is increased which is one of the reasons diversity might be improving performance. We are happy to add these observations in the discussion and motivate our future work that proposes a combination of both the diversity coefficient (~coverage) and alignment as a data quality metric.
>
> Answers to questions:
> We are happy to emphasize the remarks made in appendix A in the revisions ot our paper. In addition, we are happy to also remind the reader that a perturbation to a weight in governed (to 2-order approx) by the Fisher information matrix, thus, suggesting it is a good finger print measuring the importance of weights to a specific task. This is currently true that the vector is of the size of the parameters. We can include this in the discussion section and some ways to remedy it e.g., via dimensionality reduction + if we store the vectors we don’t have to recompute them which makes it easier to work with them.
> I think this is a correct observation that we over stated the interpretations. We are happy to adjust commensurate to what we actually did. We do hypothesize they conclusions are likely valid because the all random initialization are around the origin with some distribution related to a Gaussian, so we hypothesize our observations hold on average. We want to emphasize that the most important thing is to use a consistent probe network throughout so that diversity of coefficients are comparable.
> The main insight from using the diversity coeff. vs. cross-diversity is that our conclusions are robust to different ways one could reasonable define diversity. Which strengthens the observations of the high diversity of LLM open training data sets. The empirical relationship of diversity & cross-diversity is that the former results higher values because it does not mix sequences between data sets. So different data sets are compared directly. An advantage of cross-diversity is that it can also be used to compute the average difference between two data sets cleanly.
> Happy to fix!
> Interesting question. We first observe that the difference is only 0.002 between the two and the confidence interval for C4+WikiText is 0.001 which makes the two values much closer. An alternative analysis using Cohen’s d to measure effect size reports the difference between the two is 0.1585, while this effect size is usually considered low in classical statistics.

---

> > ### Author Response · Authors · 2023-11-20
> > **Response to great questions (cont.)**
> >
> > In addition, the cross-diversity coefficient shows the diversity is **larger** than C4, alone. We interpret all these observations together that C4 is already quite diverse, and it seems adding more data doesn’t change the diversity much and might be close to the variations bounds that might be expected when trying to make an already diverse data set more diverse.
> > Great question, we will clarify the detail in the appendix in the revision + our code will make it fully transparent where these numbers come from. The mixture proportions were computed such that they were faithful to the original mixture in LLaMAv1. For example if LLaMAv1 had a ratio of crawl data to wiki data of 2:1 then we choose our mixture proportions to respect that as much as possible. We chose LLaMAv1 because 1. We want our diversity values to be reflective of what is done in practice 2. we chose LLaMAv1 because it’s widely used and open mixture proportions are public/open (LLaMAv2 are closed & a trade secret).

---

> > > ### Comment · Reviewer_Z6o3 · 2023-11-21
> > >
> > > Thanks for the reply. I have read the rebuttal and decided to maintain my score.
> > > A kind reminder is, please consider writing the reply in a point-to-point manner, such as:
> > > Q1: ...
> > > A1: ...
> > > or so
> > >
> > > I spent quite a few minutes to figure out which question are you replying to.

---

> ### Author Response · Authors · 2023-11-22
> **Thanks for Your Consideration**
>
> Hello,
>
> Thank you for considering our response! We appreciate the time you’ve taken to consider the strengths of our paper, and respect your decision with regards to our score.
>
> We apologize for the lackluster formatting. Thus, we’ve enhanced and structured our response below as a matter of good form.
>
> ### Of course, we completely understand if your mind is made up, and, to be clear, *we have no expectation whatsoever of you reading our below responses.*
>
> (All that being said, we would of course be very grateful if you found a few minutes to give the bolded portions a skim!)

---

> ### Author Response · Authors · 2023-11-22
> **Thanks for Insightful Critique | Weaknesses 1 - Question 1**
>
> Thank you for your review of our work! We’re pleased to know that we have a shared belief in the importance of understanding the role of data quality in the development of LLMs, and that you find our introduction of the diversity coefficient as way of conceptualizing quality a merit of our paper.
>
> We also believe you raise some thoughtful points of critique, and we address them below. We fully acknowledge that our responses here are relatively long; we endeavored to respond to your insightful critiques thoroughly, and hope our responses help you better evaluate the merits of our paper.
>
> ### *Please feel free to skim the bolded sentences to get a quick overview of our main points.*
>
>
> ## Weakness 1
>
> This is an interesting point! One of the main reasons we used the theoretical lower/upper bounds is because the diversity coefficient is a completely new metric for quantifying diversity formally. Therefore, before this work there was no sense what the minimum/maximum would be for this metric.  **E.g. if the lower bound (uniformly 1 token) is 0.15 and the upper bound (random tokens) is 0.2, then a coefficient of 0.16 seems low. Alternatively, if the lower bound is 0.0 and the upper bound is 0.175, a coefficient of 0.16 seems high.**  Therefore, we verified that 1. the  **metric indeed behaves as expected at these extremes**  and 2. real world data sits at a reasonable point in relation to these extremes.
>
> In addition, we also provide  **real world, natural data reference points by analyzing 12 different real world NLP datasets. Three of these (NIH, USPTO, PUBMED) are single topic/area data sets, which are highly uniform in content and style/format and, accordingly, have relatively low diversity coefficients.**  Hence, we see that the ratio of the least diverse to most diverse datasets tested is d(Pile-cc)/d(NIH)=0.2497/0.15=1.67, which still suggests the web crawled type data are nearly twice as diverse as single-source, narrow-domain data. We are happy to  **emphasize this later point and include in the discussion**  section the nuances and importance of the upper/lower bounds in our work.
>
> ## Weakness 2
>
> We agree that this is an important limitation to mention, and will **include it in our discussion.** Though an important note is that despite using fine-tuning, we only fine-tune the final layer.
>
> ## Weakness 3
>
> We are  **happy to add more recent data-centric works to our Related Work.**  In particular, D4’s support of semantic deduplication methods appears to be of strong support and relatedness to the diversity coefficient. In addition, works like DoReMi (by Xie et. al.) also signal the importance of intelligent choice of data mixtures i.e. sub-datasets and their proportions.
>
> ## Weakness 4
>
> We are happy to  **compute diversities for pre-training data sets coding datasets.**
>
> ## Weakness 5
>
> Great observation! We are happy to address it clearly in the revision. We tried addressing it via two ways:
> 1. We used  **OpenWebText2 to avoid overlaps**  with the training set and we  **still observed the described trend (with lower CE losses).**
> 2. In Figure 6 in the appendix, we computed the “alignment coefficient” of the train and the test data and observed a positive correlation of r=0.8, meaning that the story is more nuanced as you hinted & alignment in train and test are important.
> 3. We have gathered new results, evaluating the experimental models  **on Pile-CC, which has no overlap with any training dataset. We again observe lower CE losses with higher training data diversity. We would be happy to update our figure in Appx. G with this result.**
> 4. We do want to emphasizes that we used the validation sets at eval, so although there is overlap in terms of concepts/type of data, the test data has never been seen during training. In addition, when the data is more diverse, the likelihood of training on more relevant data is increased, which is one of the reasons greater diversity likely improves performance. We are happy to  **add these observations in the discussion**  and motivate our future work that proposes a combination of both the diversity coefficient (~coverage) and alignment as a data quality metric.
>
> ## Question 1
>
> We are happy to  **emphasize the remarks made in appendix A in the revisions of our paper.**  In addition, we are happy to also remind the reader that a perturbation to a weight in governed (to 2-order approx) by the Fisher information matrix, thus, suggesting it is a  **good fingerprint measuring the importance of weights to a specific task.**  It is true that the **embedding vector is of the size of the parameters.**  We can include this in the discussion section and some ways to remedy it e.g., via dimensionality reduction + if we store the vectors we don’t have to recompute them which makes it easier to work with them.

---

> ### Author Response · Authors · 2023-11-22
> **Question 2 - Question 6**
>
> ## Question 2
>
> I think this is a  **correct observation that we over stated the interpretations. We are happy to adjust commensurate to what we actually did.**  We do hypothesize the conclusions are likely valid because all random initialization are around the origin with some distribution related to a Gaussian, so we hypothesize our observations hold on average. We want to emphasize that the most important thing is to use a consistent probe network throughout so that diversity of coefficients are comparable.
>
> ## Question 3
>
> The main insight from using the diversity coeff. vs. cross-diversity is that our conclusions are  **robust to different ways one could reasonable define diversity,**  which strengthens the observations of the high diversity of open-source LLM training datasets. The empirical relationship of diversity & cross-diversity is that the latter results in higher values because all text for a batch embedding is drawn from a _single_ sub-dataset, rather than randomly from any dataset. Hence, embeddings of different sub-datasets often push up average distance between embeddings given the clean informational separation between the batches of text.
>
> Therefore,  **cross-diversity allows different sub-datasets to be directly compared directly with one another, which can be important if one is aiming to construct corpora by using diverse _subsets_ of data**  (as is the case with The Pile). (Normal) diversity, on the otherhand, is very useful to assess the variation of a dataset when  **agnostic to or lacking information about specific sub-datasets used.**
>
> ## Question 4
>
> **Happy to fix!**
>
> ## Question 5
>
> Interesting question. We first observe that the difference is only 0.002 between the two and the confidence interval for C4+WikiText is 0.001, which means the two values are very close. An alternative analysis using Cohen’s d to measure effect size reports the difference between the two is 0.1585, and this effect size is usually considered low in classical statistics. In addition, the cross-diversity coefficient shows the diversity is greater than C4 alone, demonstrating that there is  **significant informational difference _between_ C4 and WikiText.**  Hence, we conclude that  **since C4 is already very diverse, adding in text that’s diverse from C4 (namely WikiText) doesn’t lead to the _overall_ diversity of C4+WikiText changing much,**  as the level of variation amongst batches of text (agnostic of what subset it comes from) is still just about as high after as it was before.
>
> ## Question 6
>
> Great question, we will  **clarify the detail in the appendix in the revision + our code will make it fully transparent where these numbers come from.**
>
> The mixture proportions were computed such that they were faithful to the original data mixture used for LLaMAv1. For example, if LLaMAv1 had a ratio of crawl data to wiki data of 2:1 then we choose our mixture proportions to match that as closely as possible. We chose LLaMAv1 in particular because:
> 1. We want our diversity values to be reflective of what is done in practice
> 2. We chose LLaMAv1 because it’s widely used and open mixture proportions are public/open.

---

### Official Review · Reviewer_FQiZ · 2023-10-28

**Soundness:** 1 poor
**Presentation:** 1 poor
**Contribution:** 1 poor
**Rating:** 1
**Confidence:** 4

**Summary:**

The paper focuses on the quality of data used for training Large Language Models. It introduces the Task2Vec diversity coefficient as a metric to evaluate this quality. By analyzing public datasets and conducting interpretability experiments, the authors claim that a higher diversity coefficient indicates better data quality. They suggest that this coefficient could serve as a useful tool for constructing high-quality datasets for training models.

**Strengths:**

1. **Addressing an Important Problem**: The paper tackles the significant issue of data quality metrics for large language models.
2. **Introduction of Diversity Coefficient in Context**: The paper brings the concept of a diversity coefficient into the discussion of data quality for large language models.
3. **Reference to Existing Literature**: The paper builds upon existing work, particularly on Task2Vec and Fisher Information Matrix.

**Weaknesses:**

1. **Limited Novelty and Insight**: The paper largely relies on existing methodologies, including Task2Vec diversity coefficient and latent concept analysis, and does not offer new or noteworthy findings.
2. **Issues with Model Updates and Task2Vec**: It's not clear if the model weight is updated after computing the Task2Vec embedding for each batch. This is crucial as it affects the validity of the reported results, especially those in Section 4.
3. **Unexplained Necessity for Task2Vec in Measuring Data Quality**: The paper uses token distribution as a metric for dataset diversity but fails to clarify why Task2Vec coefficients are needed instead of direct token distribution metrics. The lack of this clarification adds confusion and questions the relevance of using Task2Vec for data quality measurement.
4. **Unclear Practical Utility of Data Quality Metric**: The paper suggests the diversity coefficient as a potential data quality metric but does not empirically validate this claim. This is a significant concern, especially considering the emphasis on Task2Vec and model diversity in the paper.
5. **Methodological Ambiguities and Undefined Concepts**: The paper contains unclear statements and undefined notations, particularly in the methods section.

**Questions:**

### Major Concerns

1. **Potential Issue with Model Updates and Task2Vec**: Is the model weight updated after the Task2Vec embedding is computed for each batch? If so, doesn't this imply that the diversity should inherently be high since the information from the first batch has already been learned and therefore won't be reflected in the Task2Vec embedding for the second batch? How does this affect the importance of your results in Section 4?

2. **Task2Vec and Token Distribution: Confusion about Relevance to Data Quality**:
    - In Section 2.2.4, token distribution is used as a metric for dataset diversity. Why then is it necessary to calculate Task2Vec diversity coefficients by fine-tuning the model? How is this relevant to data quality?
    - Following from the above, Section 3.4 revisits the correlation between diversity coefficient and vocabulary size. Could you clarify why this aspect is repeatedly emphasized?

### Additional Questions
3. **Ambiguities in Section 2.1**:
    1. You mentioned "(partially) fine-tuning" the final layer of a fixed neural network to solve the "current task (or batch)". Could you clarify what you mean by "partially fine-tuning"? Also, how do you define "solving the current task (or batch)" in this context?
    2. The variable $t$ is used but not explicitly defined. Could you specify what $t$ represents and its range of possible values?
    3. When discussing the expectation over both $t$ and $\hat{x}_t$, could you elaborate on why and how this expectation is taken?

4. **Minor Formatting Issues**: The in-text citation style appears to be inconsistent across the paper.

5. **Potential Typo on Page 2**: In the contributions section, item 4 states, "...the **high** diversity of public datasets for LLM pre-training is **high**..." This appears to be a typo.

---

> ### Author Response · Authors · 2023-11-22
> **Thanks for Thorough Feedback | Weaknesses 1-4**
>
> Thank you for your review of our work! We’re pleased to know that we have a shared belief in the importance of understanding the role of data quality in the development of LLMs, and that you find our introduction of the diversity coefficient, built off strong foundations from prior work, a merit of our paper.
>
> We also believe you raise some thoughtful points of critique, and we address them below. We fully acknowledge that our responses here are relatively long; we endeavored to respond to your insightful critiques thoroughly, and hope our responses help you better evaluate the merits of our paper.
>
> ### *Please feel free to skim the bolded text to get a quick overview of our main points.*
>
> ## Weakness 1
>
> The novelty of our work is the use of previous work to  **measure important, non-trivial concepts, like the inherent diversity of language data, and make novel, quantitative, non-trivial observations of LLM pre-training datasets.**  In addition, we are the first to show with a quantitative diversity metric the  **relationship between diversity and performance (see Appendix G, in Supplementary Materials).**  Hence, we respectfully disagree that our work lacks novelty, as we believe our work is noteworthy by being the first to  **extensively test and propose a formal and rigorous diversity metric for use on real-world datasets used to train foundation models.**
>
> ## Weakness 2 (Cont. in Question 1)
>
> Briefly, we use a  **separate, deep-copied probe network for each text batch, such that the updates from finetuning on batch N are not carried over when finetuning on batch N+1.**  Hence, there is no concern of information from one batch affecting the embedding of another batch.
>
> That being said, we agree that this detail was not clearly explained in the paper, and  **will update section 2.2.3 (in Methods) to clearly state this detail.**
>
> ## Weakness 3
>
> At a high level, Task2Vec is our method of  **capturing the information of a particular sample of a dataset quantitatively;**  we then find the expected (cosine) distance between these Task2Vec embeddings to see how varied the information in a dataset is. Previous work has shown that Task2Vec embeddings appear to  **capture important characteristics of the embedded data [1].**  As such, Task2Vec provides us a much  **richer (and novel) capture of variation of text information than raw token distribution statistics.**  E.g. raw token distributions may only very coarsely detect the topic of text, and do not (explicitly) detect syntactical features like variation in writing style and document format. As explained in Question 2, we explain why, for certain lines of experimentation, we do in fact use token distribution metrics.
>
> One may then ask why we use Task2Vec embeddings, rather than the more model conventional activations. We use Task2Vec because  **activations were not successful at measuring diversity in our preliminary experiments.**  One major issue we observed was that the distances between activations were consistently very large in magnitude (we observed this behavior on both on Mini-Imagenet and on GPT-2 with random tokens when we varied the vocabulary size from 1 to whole vocabulary). We are happy to provide data and plots we have to illustrate this point.
>
> In addition, the expected cosine distance between activations was always within a small range, making the  **activations not very good at measuring variation in diversity among datasets.**
>
> Finally, the most concerning part was the large jumps in expected cosine distance we saw from small to large vocabularies,  **implying instability of an activations-based metric.**
>
> In contrast, the FIM measures which parameters are important via a second order approximation (the reason fine-tuning the final layer is important). Therefore,  **Task2Vec attempts to obtain a unique task dependent vector (in contrast to activations), and there is strong evidence that it can do so successfully [1].**
>
> ## Weakness 4
>
> First, we argue that  **it has generally been a commonly held wisdom in the LLM community that using diverse pretraining data benefits downstream model performance.**  For a few illustrative examples, consider the that:
>
> - The creators of GPT-3 made a deliberate effort to create a dataset composed of (qualitatively) diverse data [2] .
>
> - The creation of The Pile dataset was directly motivated by the desire to create an open-source high quality, diverse dataset [3]. A number of leading open-source models are partially trained on The Pile, such as Meta’s OPT models and LLaMA 1 models, Microsoft’s Megatron-Turing NLG 530B, and EleutherAI’s GPT-Neo models.
>
> - The creators of LLaMA 1 made a deliberate effort to train on diverse data, particularly preprocessed web-crawls, which they noted improved performance [4]. Note that the diversity coefficient recognizes that C4 and Pile-CC, pre-processed CommonCrawl datasets, are among the most diverse datasets tested.

---

> ### Author Response · Authors · 2023-11-22
> **Weaknesses 4 - Question 2**
>
> ### (Weakness 4 cont.)
> - The creators of the TinyStories dataset make a deliberate effort to make their data sufficiently diverse in order to improve downstream model performance [5].
>
> Given the above evidence of the importance diversity among researchers and practitioners, consider the following concrete use-cases:
>
> ### Use-case 1:  **Rigorously characterizing the effect of data diversity on (general) downstream LLM performance.**
>
> As argued in the above section, there is already a widespread belief among practitioners and researchers that training models on diverse pretraining data likely leads to better general downstream performance on language-based tasks. That being said,  **this conjecture remains highly qualitative and imprecise**  as to e.g. the optimal degree of diversity for pretraining data (e.g. TinyStories appears to indicate ideal data diversity depends on the size of the model being trained [5]).
>
> By using the diversity and cross diversity metrics, researchers can  **gain a precise, quantitative, and, in our view, well-supported measurement of the inherent variability of a model’s pretraining data. Hence, this empowers researchers to more rigorously analyze how model abilities develop with respect to this important characteristic of the data.**  For example, the diversity metric allows for a rigorous and precise study of how accuracy on a wide-ranging benchmark like MMLU varies when models are trained on more or less diverse pretraining corpora.
>
> ### Use-case 2:  **Enabling a better choice of training corpora for the training and development of an LM.**
>
> The diversity coefficient & cross diversity coefficient can serve as a  **valuable tool practitioners (and researchers) can use to curate their training corpora.**  For instance, many researchers and practitioners operate under limited compute budgets for training their LMs. When curating training corpora, scaling laws provide guidance on the optimal size of the dataset used one should use during training, but this  **leaves ambiguous what characteristics of the data one should optimize for when selecting the N billion tokens one will use**  for training. By  **comparing against the diversity values of widely used/known datasets we provide in our paper,**  the practitioner could test the diversities of candidate datasets to determine whether each is sufficiently diverse (and not overly diverse) for their purposes and choose to train on the data which satisfies their desired level of diversity.
>
> For another example of using diversity metrics for dataset curation, consider a researcher aiming to create a large natural language corpora of high quality, diverse training data. One could  **use the cross-diversity coefficient between candidate datasets to determine which datasets to include in one’s overall corpora in order to ensure one’s corpora is composed of maximally (or, rather, optimally) diverse subsets.**  In fact, the researcher or practitioner can collect small ‘sample datasets’ from certain sources (e.g. transcriptions of the most listened to podcasts on Spotify) and  **test whether this data is diverse enough from one’s existing corpora.**  Depending on if the sample data is sufficiently diverse, one could  **make an informed decision as to whether to continue scraping**  the given source, or search for a more diverse source.
>
> ## Weakness 5
>
> We clarify the ambiguities in Question 3, and will  **happily update our writing in methods to reflect greater clarity of these concepts.**
>
> ## Question 1
>
> ### Potential Issue with Model Updates and Task2Vec
> Great question! We use a  **separate, deep-copied probe network for each text batch, such that the updates from finetuning on batch N are not carried over when finetuning on batch N+1.**  Hence, there is  **no concern of information from one batch affecting the embedding of another batch.**  We are happy to update our section 2.2.3 to clarify this detail.
>
> ###  If so, doesn't this imply…
> We believe the above point answers this concern. In addition, if this were an issue, it would have showed up in our sanity check when computing the theoretical lower bound of the diversity coefficient, i.e. when the data comes from a distribution with almost all probability mass on one token.  **Since the diversity coefficient for this lower bound is ~0.05, very close to the minimum of zero, the evidence likewise suggests we don’t have this problem.**
>
> ## Question 2
>
> ### In Section 2.2.4…
> **Task2Vec uses the diagonal of the FIM to compute a “fingerprint” of a task. The FIM can only indicate which weights are important in the new task if some of the weights change.**  This is because the FIM is a 2nd order approximation to a perturbation in the weights i.e., E_{x ~\sim \hat{p}}[KL(p_w' || p_w)] = \delta_w F \delta_w^\top + o(delta(w)^2). Therefore, the theory suggests we need  **a change in the direction of the relevant task (or current batch) to get the relevant vector/embedding for that task/batch.**

---

> ### Author Response · Authors · 2023-11-22
> **Question 2 - Question 5 | References**
>
> ### (Question 2 cont.)
>
> Hence, Task2Vec fine-tunes the final layer on the text batch for the task of next-token prediction on the given text sequences.
>
> Intuitively, diverse data is an important factor in preventing model overfitting (e.g. LLMs trained on only dialog-format data would have great difficulty writing an argumentative essay) and  **promoting generality of model ability,**  as the model has been trained to perform well on a wide variety of tasks (i.e. next-token prediction on a wide variety of text). Furthermore, using diverse training data increases the chance that test-time demands are similar to data seen during training,  **providing the model something akin to ‘in-domain’ performance for a larger number of domains.**
>
> In this instance, we create synthetic data using extreme token distributions to examine the practical upper & lower bounds that our metric can attain and thereby  **give context to the coefficients of real world datasets.**
>
> ### Section 3.4 revisits…
> To validate the diversity coefficient is a reliable metric for diversity, it has to behave in ways we’d expect a diversity metric would. By including  **experiments on the size of the vocabulary, which is a controlled & clearly interpretable test setting, we show that the diversity coefficient recognizes when data uses a richer vocabularly, i.e. is clearly more diverse.**
>
> Thus, we argue this is evidence that the diversity coefficient is also  **able to recognize this type of richness & diversity of data in cases of natural language corpora,  where the size of vocabulary (or number of latent concepts) in untenable to precisely define.** Furthermore, by experimenting on variation in the vocabulary size, we show that  **the diversity coefficient captures a broad notion of data diversity,**  i.e. not simply variation in number of latent concepts. We are happy to clarify this motivation in our revisions, if you find it would add to the strength of our paper.
>
> ## Question 3
>
> ### 3.1
> By “partially fine-tune the final layer,” we mean we  **do a limited number of updates with Adam to the final layer of the probe network using the current batch.**  This is required by the Task2Vec method.
>
> By “solving the current task,” we mean  **“minimize the cross-entropy loss**  on the next token prediction task on the given batch of text sequences”.
>
> ### 3.2
> **$t$ is the index for the current token and, thus, $t$ ranges from 0 to the length of sequence.**  We use this notation to indicate that we take the average across each token of the sequence when computing log loss.
>
> ### 3.3
> The FIM is the  **expected covariance of the scores**  (gradients of the log-likelihood) with respect to the model parameters. Therefore, to calculate the FIM, we find the  **average log likelihood of predicted tokens** $\hat{x}\_t$ **given context** $x\_{1:t-1}$ **across indices** $t$ **in sequence** $x$ **(conventional log-likelihood in NLP), find the covariance matrix of this loss wrt the probe network parameters** $w$, **and then find the expectation of these covariance matrices across indices** $t$ **in sequence** $x$ **and sequences** $x$ **from batch** $B$.
>
> We agree that some of these details may be confusing as currently presented in the paper, and **we would be happy to add clarifications of 3.1, 3.2, 3.3 to the paper.**
>
> ## Questions 4 & 5
>
> **We’re happy to fix citation formatting errors and typos in the paper.**
>
> ## References
>
> [1] Achille et. al. Task2Vec: Task embedding for meta-learning. 2019. https://arxiv.org/pdf/1902.03545.
> Key Quote: “The FIM is also related to the (Kolmogorov) complexity of a task, a property that can be used to define a computable metric of the learning distance between tasks.”
>
> [2] Brown et. al. Language models are few-shot learners. CoRR, 2020. https://arxiv.org/abs/2005.14165.
> Key Quote: “Our basic pre-training approach…[includes] scaling up of dataset… diversity”
>
> [3] Leo Gao et. al. The pile: An 800gb dataset of diverse text for language modeling. 2020. https://arxiv.org/abs/2101.00027.
> Key Quote: “Recent work has demonstrated that increased training dataset diversity improves general cross-domain knowledge and downstream generalization capability for [LLMs]. With this in mind, we present The Pile…”
>
> [4] Touvron et, al. Llama: Open and efficient foundation language models. 2023. https://arxiv.org/abs/2302.13971.
> Key Quotes: “Our training dataset is a mixture of several sources…that cover a diverse set of domains”; “During exploratory experiments, we observed that using diverse pre-processed CommonCrawl datasets improves performance”
>
> [5] Eldan et. al. TinyStories: How small can language models be and still speak coherent english?. 2023. https://arxiv.org/abs/2305.07759.
> Key Quote: “The main challenge in using [LLMs] for producing training data is generating a dataset that is sufficiently diverse…”

---

> > ### Comment · Reviewer_FQiZ · 2023-11-23
> > **Thank you for your response**
> >
> > I would like to thank the authors for providing this long response, though my concerns and questions cover the same points so you could have answered them once.
> >
> > My question about the model updates was clarified by the authors. Thank you.
> >
> > Regarding the necessity of Task2Vec in measuring data quality, the authors responded that "_richer (and novel) capture of variation of text information than raw token distribution statistics_," but I don't see this claim supported by any experiment or theory in the paper, especially regarding its advantage in measuring data quality.
> >
> > Regarding my concern about the practical utility of the proposed diversity coefficient, I am aware that diversity is important for data curation. While I can imagine all these scenarios the authors mentioned, I was mainly asking for empirical evidence of the effectiveness of using the proposed metric in any of these scenarios, which is missing from the paper and not provided in the response.
> >
> > Overall, while I appreciate the authors' efforts in writing this long response, my main concerns were not adequately addressed. Therefore, I would like to keep my original rating.

---

### Official Review · Reviewer_H6fT · 2023-10-30

**Soundness:** 4 excellent
**Presentation:** 3 good
**Contribution:** 3 good
**Rating:** 6
**Confidence:** 2

**Summary:**

This paper begins by observing that current trends in pre-trained large language models (LLMs) mostly focus on scaling model and data size, while ignoring the quality of pre-training data. The author proposes that the quality of pre-training data is also an important factor for training powerful LLMs. To this end, they propose using the recently proposed Task2Vec diversity coefficient to ground and understand formal aspects of data quality. The authors also validate the diversity coefficient by demonstrating its interpretability and correlation with intuitive diversity properties aligned with human intuitions, and provide ways to evaluate the diversity coefficient for a given dataset.

**Strengths:**

1. This paper is well-written and rigorous.
2. I find the idea of improving the performance of pre-trained large language models from a data quality perspective meaningful. Considering the increasing pre-training cost, using small and high-quality data and avoiding the scaling of model and data size is a good way.
3. The authors demonstrate that the public datasets for LLM pre-training have high formal diversity using well-motivated lower and upper bounds. They also provide practitioners with simpler ways to evaluate the diversity coefficient. It is more important.

**Weaknesses:**

1. The definitions in sections 2.1 and 2.2 are essential to understanding the paper’s main argument. However, they are somewhat difficult to comprehend. To make these definitions more accessible, I suggest providing more illustrations in addition to citing related papers.

2. More experiments should be conducted to investigate the correlation between diversity coefficient and LLM performance. For example, on more models and datasets with multiple runs. The end goal of this research should be to improve the performance of LLMs.

**Questions:**

N/A

---

> ### Author Response · Authors · 2023-11-22
> **Thanks for Insightful Feedback + Weaknesses 1-2**
>
> Hello,
>
> Thank you for your review of our work! We’re pleased to know that you appreciate the writing quality of the paper, and that we have a shared belief in the importance of using a data quality-centric perspective when thinking about training LLMs. In addition, we’re glad that you find our analysis of the diversity of public datasets and practical methods for use of our metric compelling!
>
> We also believe you raise some thoughtful points of critique, and we address them below.
>
> ### *Please feel free to skim the bolded sentences to get a quick overview of our main points.*
>
>
> ## Clarity of Definitions in Methods Section
>
> We agree that this portion of the paper is fairly dense at the moment, and that clearer explanations, visual aids, and citing similar work would help better communicate this important mathematical framework.
>
> In terms of visual aids,  **we include an illustration of the diversity coefficient computation pipeline in Appx. F, Figure 5.**  Unfortunately, we only reference this visual aid near the end of Section 2.2, and do not do so with sufficient emphasis. Hence, we will  **revise Sections 2.1 and/or 2.2 to include an earlier and/or emphasized reference to our illustration of the diversity computation pipeline**  so that it may serve as an effective visual aid for understanding Sections 2.1 and 2.2.
>
> In terms of related works, we also agree that adding in references to past related work would help shore up one’s understanding of our approach. As such,  **we will edit Sections 2.1 and 2.2 to include helpful references to related work,**  such as Edwards et. al. (https://arxiv.org/pdf/1606.02185), a precursor of Task2Vec which similarly uses an unsupervised method of training a probe network on a dataset in order to extract meaningful quantitative information about the characteristics of the dataset, and Ngyuyen et. al., (https://arxiv.org/pdf/1908.01091) which also uses Task2Vec embeddings to determine properties of a certain data sequence (in this case, the ‘hardness’ of the sequence in a continual learning setting).
>
> We will also  **generally proofread and revise this section for clarity**  so that our methods are more readily understandable. If you have any other ideas for specific improvements, or believe our above proposed changes may be inadquate, please feel free to suggest!
>
>
> ## Experiments on the Relationship Between the Diversity Coefficient and LLM Performance
>
> We agree that this is an important area for investigation! As such, we have conducted preliminary experiments on models of small scale (using 50M parameter models, trained on ~1.3B tokens; see Appendix G, within Supplementatry Material, for details) and  **found that models trained on more diverse data achieve lower loss (i.e. higher performance) on diverse eval datasets.**  This indicates that a high diversity coefficient of pretraining data lead to a meaningful in downstream model performance.
>
> We also find that  **it is generally a commonly held piece of wisdom in the LLM community that using diverse pretraining data benefits downstream model performance:**
>
> - The creators of GPT-3 made a deliberate effort to create a dataset composed of (qualitatively) diverse data [1] .
>
> - The creation of The Pile dataset was directly motivated by the desire to create an open-source high quality, diverse dataset [2]. A number of leading open-source models are partially trained on The Pile, such as Meta’s OPT models and LLaMA 1 models, Microsoft’s Megatron-Turing NLG 530B, and EleutherAI’s GPT-Neo models.
>
> - The creators of LLaMA 1 made a deliberate effort to train on diverse data, particularly preprocessed web-crawls, which they noted improved performance [3]. Note that the diversity coefficient recognizes that C4 and Pile-CC, pre-processed CommonCrawl datasets, are among the most diverse datasets tested.
>
> - The creators of the TinyStories dataset make a deliberate effort to make their data sufficiently diverse in order to improve downstream model performance [4].
>
> Given we believe the diversity coefficient closely tracks an intuitive human sense of how diverse data is, the above points present further evidence that using  **training data with a sufficiently high diversity coefficient is likely to lead to better downstream performance.**

---

> ### Author Response · Authors · 2023-11-22
> **Weakness 2 cont. | References**
>
> ### (Weakness 2 cont.)
>
> In addition, we observe  **compelling empirical evidence that diverse pretraining data is an important driver of In-Context Learning (ICL) in LLMs:**
>
> - A ‘mixture of concepts’ structure in pretraining data is key for instilling ICL ability, and data with only 1 concept i.e. data with very low semantic and syntactic diversity failed to produce ICL in transformer models [5]. Note that Section 3.4 demonstrates that the diversity coefficient of data strongly correlates with the number of latent concepts in (synthetic) data.
>
> - Training models on qualitatively diverse natural language corpora strengthened models’ ICL ability, such as using training corpora composed of intuitively cross-diverse sub-datasets [6].
>
> - Training transformers on data with a large number of rarely occurring classes (in this case, character symbols), and with dynamic item meanings and interpretations lead to higher ICL ability [7].
>
> Though our main goal of our paper is to establish and support the diversity coefficient as a measure of data diversity (and, thus, we leave more significant experimentation on the relationship between the diversity coefficient and performance to future work), we still agree that we did not address this topic sufficiently compellingly in our introduction, results, and/or discussion sections.
>
> Hence, if you believe it would contribute to the strength of our paper, we would be happy to:
>
> - **Edit and revise our introduction and/or discussion**  to better reflect our understanding of the connection between data diversity and model performance.
>
> - **Expand our results from appendix G by conducting a similar experiment with ~125M parameter models trained on ~2.5B tokens**  (using ~3-4 pretraining datasets that vary in diversity), and evaluating the downstream performance of these models on diverse text.
>
>
> ## References
>
> [1] Brown et. al. Language models are few-shot learners. CoRR, 2020. https://arxiv.org/abs/2005.14165.
> Key Quote: “Our basic pre-training approach…[includes] scaling up of dataset… diversity”
>
> [2] Leo Gao et. al. The pile: An 800gb dataset of diverse text for language modeling. 2020. https://arxiv.org/abs/2101.00027.
> Key Quote: “Recent work has demonstrated that increased training dataset diversity improves general cross-domain knowledge and downstream generalization capability for large-scale language models. With this in mind, we present The Pile…”
>
> [3] Touvron et, al. Llama: Open and efficient foundation language models. 2023. https://arxiv.org/abs/2302.13971.
> Key Quotes: “Our training dataset is a mixture of several sources…that cover a diverse set of domains”; “During exploratory experiments, we observed that using diverse pre-processed CommonCrawl datasets improves performance”
>
> [4] Eldan et. al. TinyStories: How small can language models be and still speak coherent english?. 2023. https://arxiv.org/abs/2305.07759.
> Key Quote: “The main challenge in using large language models for producing training data is generating a dataset that is sufficiently diverse…”
>
> [5] Xie et. al. An explanation of in-context learning as implicit bayesian inference. 2021. https://arxiv.org/abs/2111.02080
> Key Quote: “When pretrained with only one concept, in-context learning fails.”
>
> [6] Shin et. al. On the effect of pretraining corpora on in-context learning by a large-scale language model. 2022. https://arxiv.org/pdf/2204.13509.
> Key Quote: “Our study shows that diverse pretraining corpora strengthen the ability of in-context learning.”
>
> [7] Chan et. al. Data distributional properties drive emergent in-context learning in transformers. 2022. https://arxiv.org/pdf/2205.05055.
> Key Quote: “[Data with] a large number of rarely occurring classes [promotes in-context learning]”

---

> > ### Comment · Reviewer_H6fT · 2023-11-22
> > **To authors**
> >
> > Thanks for the reply. My concerns are addressed. I will keep my positive rating.

---

> > > ### Author Response · Authors · 2023-11-22
> > > **Thanks for Your Consideration**
> > >
> > > Hello,
> > >
> > > Thank you for taking the time to consider our response! We're glad we've been able to address your concerns, and are grateful for your decision with respect to our score.

---

### Official Review · Reviewer_kREU · 2023-11-02

**Soundness:** 2 fair
**Presentation:** 3 good
**Contribution:** 2 fair
**Rating:** 3
**Confidence:** 4

**Summary:**

The paper extends the diversity coefficient metric [1] to textual pre-training datasets. To be more specific, the approach first estimates the Task2Vec embeddings [2] for batches of sequences in the dataset, followed by computing the expected distance between two random batches. This diversity metric is then computed on various existing pre-training datasets (and their combination), followed by some analysis of the same.

[1] Brando Miranda, Patrick Yu, Yu-Xiong Wang, and Sanmi Koyejo. The Curse of Low Task Diversity: On the Failure of Transfer Learning to Outperform MAML and Their Empirical Equivalence.

[2] Alessandro Achille, Michael Lam, Rahul Tewari, Avinash Ravichandran, Subhransu Maji, Charless C. Fowlkes, Stefano Soatto, and Pietro Perona. Task2vec: Task embedding for meta-learning.

**Strengths:**

- The motivation behind building data quality metrics is an important direction for building better language models.
- The paper is well written and easy to follow along.

**Weaknesses:**

- Unclear practicality of the proposed metric: while the metric is shown to reasonably estimate the inherent “diversity in a dataset,” I’m quite unsure about its extrapolation to measuring “dataset quality.”
- Limited novelty: The paper directly builds on two existing lines of work [1, 2], and is also limited by its application to different practical scenarios (more in questions).
- The proposed diversity metric is model-dependent (GPT-2 is used in the paper), while the downstream effect of the probe model on the diversity metric isn’t studied (more in questions).

[1] Brando Miranda, Patrick Yu, Yu-Xiong Wang, and Sanmi Koyejo. The Curse of Low Task Diversity: On the Failure of Transfer Learning to Outperform MAML and Their Empirical Equivalence.

[2] Alessandro Achille, Michael Lam, Rahul Tewari, Avinash Ravichandran, Subhransu Maji, Charless C. Fowlkes, Stefano Soatto, and Pietro Perona. Task2vec: Task embedding for meta-learning.

**Questions:**

- What are some practical use-cases for the proposed diversity metric?
- Keeping aside the vague intuition, why do you think data diversity should be a good indicator of data quality?
- The considered scenario for estimating the utility of a larger batch-size is also debatable: a larger batch-size in addition to providing diverse data also directly affects the optimization. Nonetheless, in addition to showing a saturating diversity with increasing batch size, how does the model performance change with increasing batch size?
- Assuming data diversity is a good indicator of data quality, do you think data diversity estimated using GPT-2 as the probe-model would translate to better training of (1) models with different architectures (e.g., encoder-decoder, encoder-only), and (2) different model-sizes (e.g., GPT-3, …)?

---

> ### Author Response · Authors · 2023-11-22
> **Thanks for Insightful Critique + Response Overview**
>
> Hello,
>
> Thank you for your review of our work! We’re pleased to know that you appreciate the quality of writing in the paper, and that we have a shared belief in the importance of understanding the role of data quality in the performance and development of LLMs.
>
> In addition, we believe you raise some thoughtful points, and we address them below. We organize our responses by section (denoted by “ # ”), where each section addresses a specific point or set of points you raise.
>
> 1. Our first comment below addresses the connection between data diversity and data quality.
> 2. Our second comment below addresses practical use-cases of the metric.
> 3. Finally, our third comment addresses the implications of the diversity coefficient for models of varied size and architecture and the novelty of our work. It also contains a basic reference list, which we refer to throughout our response.
>
> We fully acknowledge that our responses here are relatively long; we endeavored to respond to your insightful critiques thoroughly, and hope our responses help you better evaluate the merits of our paper.
>
> To assist with ease of reading and understanding, we have bolded our key points, so
>
> ### *please feel free to skim bolded sentences to quickly become acquainted with the main points of our response.*

---

> ### Author Response · Authors · 2023-11-22
> **Connection between Data Diversity and Data Quality**
>
> # Connection between Data Diversity and Data Quality
>
> We believe that the diversity of a dataset is an important determinant of data quality and a driver of increased downstream performance of models trained on these datasets, and in this section, we will provide justification of this claim.
>
> One reason for believing this is that  **it has generally been a commonly held wisdom in the LLM community that using diverse pretraining data benefits downstream model performance.**  For a few illustrative examples, consider the that:
>
> - The creators of GPT-3 made a deliberate effort to create a dataset composed of (qualitatively) diverse data [1] .
>
> - The creation of The Pile dataset was directly motivated by the desire to create an open-source high quality, diverse dataset [2]. A number of leading open-source models are partially trained on The Pile, such as Meta’s OPT models and LLaMA 1 models, Microsoft’s Megatron-Turing NLG 530B, and EleutherAI’s GPT-Neo models.
>
> - The creators of LLaMA 1 made a deliberate effort to train on diverse data, particularly preprocessed web-crawls, which they noted improved performance [3]. Note that the diversity coefficient recognizes that C4 and Pile-CC, pre-processed CommonCrawl datasets, are among the most diverse datasets tested.
>
> - The creators of the TinyStories dataset make a deliberate effort to make their data sufficiently diverse in order to improve downstream model performance [4].
>
> Thus, we see that many real-world practitioners and members of the LLM research community find that  **the diversity of pretraining data is an important driver of downstream model performance**  and, thus, that data diversity is likely a key component of data quality.
>
> Another important reason data diversity is a likely driver of data quality is that there is  **compelling empirical evidence that diverse pretraining data is an important driver of In-Context Learning (ICL) in LLMs.**  For support of this, consider the following examples:
>
> - A ‘mixture of concepts’ structure in pretraining data is key for instilling ICL ability, and data with only 1 concept i.e. data with very low semantic and syntactic diversity failed to produce ICL in transformer models [5]. Note that Section 3.4 demonstrates that the diversity coefficient of data strongly correlates with the number of latent concepts in (synthetic) data.
>
> - Training models on qualitatively diverse natural language corpora strengthened models’ ICL ability, such as using training corpora composed of intuitively cross-diverse sub-datasets [6].
>
> - Training transformers on data with a large number of rarely occurring classes (in this case, character symbols), and with dynamic item meanings and interpretations lead to higher ICL ability [7].
>
> Thus,  **dataset diversity appears to be an important driver of ICL ability, and we believe the diversity coefficient quantifies this characteristic of data well, such as its ability to recognize data with a variety of concepts/classes, and data that is intuitively/qualitatively diverse.**  Since ICL is an important factor in the overal performance and capabilities of LLMs, we believe that this is another dimension on which data diversity is an important driver of data quality.
>
> Finally, we have conducted small scale experiments (see Appendix G, within Supplementatry Material) and found that  **models trained on more diverse data achieve lower loss (i.e. higher performance) on diverse eval datasets.**  This indicates that diverse pretraining data can be a meaningful contributor to model performance.
>
> That being said, we believe  **the relationship between data diversity and data quality is likely non-monotonic.**  There is probably an optimal range for the diversity of training data for a given LLM, where data with too low of a diversity coefficient is too homogenous to ‘properly challenge’ the model (e.g. prevent overfitting to certain styles of writing) while data with too high of a diversity coefficient is too ‘chaotic’ and varied for the model to ‘learn from properly.’
>
> Therefore, we believe the diversity coefficient would allow the research community to **better assess the data quality of their training corpora with regard to whether the data is sufficiently (and not overly) diverse, enabling them to achieve their desired downstream performance.**
>
> However, we also agree that we did not explain this rationale sufficiently compellingly in our introduction, results, and/or discussion sections. Hence, if you believe it would contribute to the strength of our paper, we would be happy to:
>
> - **Edit and revise our introduction and/or discussion** to better reflect our understanding of the connection between data diversity and data quality.
>
> - **Expand our results from appendix G by conducting a similar experiment with ~125M parameter models trained on ~2.5B tokens** (using training datasets that vary in diversity), and evaluating the downstream performance of these models on diverse eval datasets.

---

> ### Author Response · Authors · 2023-11-22
> **Practical Use-Cases of the Diversity Coefficient**
>
> # Practical Use-Cases of the Diversity Coefficient
>
> Building on the reasoning from above, we present concrete practical use cases for the diversity and cross diversity metrics we propose in the paper.
>
> ## Use-case 1: **Rigorously characterizing the effect of data diversity on (general) downstream LLM performance.**
>
> As argued in the above section, there is already a widespread belief among practitioners and researchers that training models on diverse pretraining data likely leads to better general downstream performance on language-based tasks. That being said,  **this conjecture remains highly qualitative and imprecise**  as to e.g. the magitude of the effect of using diverse pretraining data and the optimal degree of diversity for pretraining data (e.g. TinyStories appears to indicate ideal data diversity depends on the size of the model being trained [7]).
>
> By using the diversity and cross diversity metrics, researchers can  **gain a precise, quantitative, and, in our view, well-supported measurement of the inherent variability of a model’s pretraining data. Hence, this empowers researchers to more rigorously analyze how model abilities develop with respect to this important characteristic of the data,**  without being impeded by the fact that real-world natural language corpora do not have clear and precise labeling of e.g. the number of unique concepts they contain. For example, the diversity metric allows for a rigorous and precise study of how accuracy on a wide-ranging benchmark like MMLU varies when models are trained on more or less diverse pretraining corpora.
>
> ## Use-case 2:  **Rigorously studying the effect of data diversity on the development of ICL ability in LLMs in more realistic and meaningful settings.**
>
> As argued in the first section, we observe experimental evidence that training data with a diversity of concepts and style/format appear to be a strong driver of ICL ability in transformers. But, these experiments were  **conducted in primarily ‘toy’ settings,**  with models 2-3 orders of magnitude smaller than current leading LLMs in terms of both parameter count and dataset size, and which use data that is drawn from synthetic rather than real-world, natural language sources. Hence, these experiments  **leave open the question of whether their results will generalize to more realistic settings.**
>
> Given we believe the **diversity coefficient allows for a well-supported, quantitative assessment of data diversity in real-world, natural language data settings,**  we believe this will empower researchers to study the emergence of ICL (with respect to data characteristics) in more realistic settings, leveraging large natural language corpora to train larger and more representative experimental models. Ultimately, this enables results such as  **finding the R^2 value of the relationship between model performance and natural language training data diversity.**  This enables researchers to more rigorously understand the mechanism by which advanced LLMs, such as LLaMA 2 or GPT-3, acquire powerful ICL abilities.
>
> ## Use-case 3:  **Enabling a better choice of training corpora for the training and development of a LM.**
>
> The diversity coefficient and cross diversity coefficient can serve as a  **valuable tool that practitioners (and researchers) can use to curate their training corpora.**  For instance, many researchers and practitioners operate under limited compute budgets for training their LMs. When curating training corpora, scaling laws provide guidance on the optimal size of the dataset used one should use during training, but this  **leaves ambiguous what characteristics of the data one should optimize for when selecting the N billion tokens one will use**  for training. By  **comparing against the diversity values of widely used/known datasets we provide in our paper,**  the practitioner could test the diversities of candidate datasets to determine whether each is sufficiently diverse (and not overly diverse) for their purposes and choose to train on the data which satisfies their desired level of diversity.
>
> For another example of using diversity metrics for dataset curation, consider a researcher aiming to create a large natural language corpora of high quality, diverse training data (similar to the intent behind The Pile). One could  **use the cross-diversity coefficient between candidate datasets to determine which datasets to include in one’s overall corpora in order to ensure one’s corpora is composed of maximally (or, rather, optimally) diverse subsets.**  In fact, the researcher or practitioner can collect small ‘sample datasets’ from certain sources (e.g. transcriptions of the most listened to podcasts on Spotify) and  **test whether this data is diverse enough from one’s existing corpora.**  Depending on if the sample data is sufficiently diverse, one could  **make an informed decision as to whether to continue scraping**  the given source, or search for a more diverse source.

---

> ### Author Response · Authors · 2023-11-22
> **Relevance of Div to Various Sizes/Archs | Novelty | References**
>
> # Relevance of Diversity Coefficient to Models of Various Sizes & Architectures.
>
> We believe that the diversity coefficient of a model’s training data has important implications for models of various architectures and/or sizes, but, as we describe below,  **what _exactly_ these implications are depend on what specific downstream model is used.**
>
> For instance, examining the TinyStories dataset, we see datasets with a level of diversity that is helpful for large models (e.g. OpenWebText or C4 for models like GPT-3 or LLaMA 1 20B) can inhibit the abilities of small models, which perform qualitatively better when trained on less (but still sufficiently) diverse text [4]. Thus, we see that  **the optimal range of diversity likely increases as the model becomes larger.**
>
> Another approach is to consider the conceptual foundations of Task2Vec embeddings. The Task2Vec method is based on the Fisher Information Matrix (FIM), which is related to the  **Kolmogorov complexity of (in our case) next token prediction on a batch of text sequences [8].**  Given the complexity of a next token prediction task is independent of what downstream model is being used, the resulting FIM, Task2Vec embedding, and, thus,  **diversity coefficient is likewise meaningful, regardless of what downstream model is being used to train on the data.**  We simply use GPT-2 as our probe network as a  **practical, reasonable, and relatively efficient implementation of the FIM computation**  need to generate a Task2Vec embedding. Given this understanding, we believe that the diversity coefficient, computed using a GPT-2 probe network, still has important implications for model performance, regardless of a model’s specific architecture (e.g. encoder only vs. decoder only).
>
> Thus, for the same reason we believe data diversity is an important factor of overall data quality, we believe that the diversity coefficient of training data has important implications for models of varied size and architecture,  **independent of the downstream model’s similarity to the particular probe network we use**  (in this case, GPT-2).
>
>
> # Novelty of our work
>
> The novelty of our work is the use of previous work to  **measure important, non-trivial concepts, like the inherent diversity of language data, and make novel, quantitative, non-trivial observations of LLM pre-training datasets.**  In addition, we are the first to show with a quantitative diversity metric the  **relationship between diversity and performance (see Appendix G, in Supplementary Materials).**  Hence, we respectfully disagree that our work lacks novelty, as we believe our work is noteworthy by being the first to  **extensively test and propose a formal and rigorous diversity metric for use on real-world datasets used to train foundation models.**
>
>
> # References
>
> [1] Brown et. al. Language models are few-shot learners. CoRR, 2020. https://arxiv.org/abs/2005.14165.
> Key Quote: “Our basic pre-training approach…[includes] scaling up of dataset… diversity”
>
> [2] Leo Gao et. al. The pile: An 800gb dataset of diverse text for language modeling. 2020. https://arxiv.org/abs/2101.00027.
> Key Quote: “Recent work has demonstrated that increased training dataset diversity improves general cross-domain knowledge and downstream generalization capability for large-scale language models. With this in mind, we present The Pile…”
>
> [3] Touvron et, al. Llama: Open and efficient foundation language models. 2023. https://arxiv.org/abs/2302.13971.
> Key Quotes: “Our training dataset is a mixture of several sources…that cover a diverse set of domains”; “During exploratory experiments, we observed that using diverse pre-processed CommonCrawl datasets improves performance”
>
> [4] Eldan et. al. TinyStories: How small can language models be and still speak coherent english?. 2023. https://arxiv.org/abs/2305.07759.
> Key Quote: “The main challenge in using large language models for producing training data is generating a dataset that is sufficiently diverse…”
>
> [5] Xie et. al. An explanation of in-context learning as implicit bayesian inference. 2021. https://arxiv.org/abs/2111.02080
> Key Quote: “When pretrained with only one concept, in-context learning fails.”
>
> [6] Shin et. al. On the effect of pretraining corpora on in-context learning by a large-scale language model. 2022. https://arxiv.org/pdf/2204.13509.
> Key Quote: “Our study shows that diverse pretraining corpora strengthen the ability of in-context learning.”
>
> [7] Chan et. al. Data distributional properties drive emergent in-context learning in transformers. 2022. https://arxiv.org/pdf/2205.05055.
> Key Quote: “[Data with] a large number of rarely occurring classes [promotes in-context learning]”
>
> [8] Achille et. al. Task2Vec: Task embedding for meta-learning. 2019. https://arxiv.org/pdf/1902.03545.
> Key Quote: “The FIM is also related to the (Kolmogorov) complexity of a task, a property that can be used to define a computable metric of the learning distance between tasks.”

---

> > ### Comment · Reviewer_kREU · 2023-11-23
> > **Response to rebuttal**
> >
> > Thanks for your elaborate responses, and highlighting the key messages in each paragraph. However, reviewing being a subjective matter, I would like to stick to my original rating.

---

### Author Response · Authors · 2023-11-23

Hello,

Thank you to all reviewers for the time and effort you’ve dedicated to help our paper be the best it can be! **We’ve found your feedback really valuable, and will work to incorporate it into our thinking and our paper.**

### As such, this is an update detailing the major additions to our most recent revised submission:
- Added references to similar works in methods to help clarify our diversity computation approach | Section 2 (Methods) | Reviewer 2 (H6fT)
- Clarified why we use Task2Vec embeddings | Section 2 (Methods) | Reviewers 1, 2, 3, 4 (All)
- Made clear reference to visual of diversity computation | Section 2 (Methods) | Reviewer 2 (H6fT)
- Clarified data mixes used, particularly Mix2 | Section 3 (Experiments), Appendix I.6 | Reviewer 4 (Z6o3)
- Acknowledged ambiguity in results of random probe network | Section 4 (Using the Div. Coeff.) | Reviewer 4 (Z6o3)
- Added recent data-quality-centric related work | Section 5 (Related work) | Reviewer 4 (Z6o3)
- Addressed limitation of computational overhead | Section 6 (Discussion) | Reviewer 4 (Z6o3)
- Fixes of some minor typos, etc. | Whole paper | Reviewers 1, 2, 3, 4 (All)
- Elaboration in Appendix A on why we use Task2Vec | Appx. A | Reviewers 1, 2, 3, 4 (All)
- Updated Appendix G Table 2 with new benchmark without training data overlap. Also corrected details and results, as we discovered a bug which lead to our previous models being trained for a differing number of tokens. | Appx. G | Reviewers 1, 2, 3, 4 (All)
- Updated Appendix H Fig 6 with corrected details and results as we discovered a bug which lead to our previous models being trained for a differing number of tokens. | Appx. H | Reviewers 1, 2, 3, 4 (All)

---

### Meta-Review · Area_Chair_4697 · 2023-12-06

**Metareview:**

The paper provides a data quality metric for LLM training data measuring the diversity of the training set. The reviews agree that the problem of evaluating the quality of a dataset is highly motivated. They also mentioned the paper to be well written and easy to follow. A major issue raised in the reviews was around the papers novelty (e.g. FQiZ: “The paper largely relies on existing methodologies, including Task2Vec diversity coefficient and latent concept analysis, and does not offer new or noteworthy findings.”, kREU: “The paper directly builds on two existing lines of work [1, 2], and is also limited by its application to different practical scenarios”). Another notable issue raised was the need to survey more recent data-centric works (see Z6o3’s comment and suggested paper). It could be that the novelty issues are a matter of a clearer comparison with previous works and a better explanation of the innovation on top of them, but either way the paper in its current form does not seem to be ready to be published in ICLR.

**Justification For Why Not Higher Score:**

Serious concerns were raised regarding the papers novelty

**Justification For Why Not Lower Score:**

n/a

---

### Decision · Program_Chairs · 2024-01-16

Reject